# Intervention and Conditioning in Causal Bayesian Networks

**Sainyam Galhotra**
Computer Science Dept.
Cornell University
sg@cs.cornell.edu

**Joseph Y. Halpern**
Computer Science Dept.
Cornell University
halpern@cs.cornell.edu

## Abstract

Causal models are crucial for understanding complex systems and identifying causal relationships among variables. Even though causal models are extremely popular, conditional probability calculation of formulas involving interventions pose significant challenges. In case of Causal Bayesian Networks (CBNs), Pearl assumes autonomy of mechanisms that determine interventions to calculate a range of probabilities. We show that by making simple yet often realistic independence assumptions, it is possible to uniquely estimate the probability of an interventional formula (including the well-studied notions of probability of sufficiency and necessity). We discuss when these assumptions are appropriate. Importantly, in many cases of interest, when the assumptions are appropriate, these probability estimates can be evaluated using observational data, which carries immense significance in scenarios where conducting experiments is impractical or unfeasible.

## 1 Introduction

Causal models play a pivotal role in elucidating the causal relationships among variables. These models facilitate a principled approach to understanding how various factors interact and influence each other in complex systems. For instance, in epidemiology, causal models often help us understand the relationship between lifestyle choices and health outcomes (Greenland, Pearl, and Robins 1999); and in economics, they help to analyze the impact of policy changes on market dynamics (Hicks 1979). These examples underscore the versatility and utility of causal models for providing a formal representation of system variables.

Interventions and conditioning are the most fundamental procedures in the application of causal models, useful to examine and analyze causal mechanisms. For example, interventions help explain the outcome of complex ML systems (Galhotra, Pradhan, and Salimi 2021); and in AI-driven healthcare diagnostics, it is crucial to discern the effect of a particular intervention (like a change in treatment protocol) on patient outcomes (Greenland 1999).

Despite their utility, calculating the probabilities related to interventions and conditioning in tandem presents significant challenges. Indeed, it is not even clear what the semantics of queries involving counterfactuals is. Work in the AI literature has focused on two types of models: *functional* causal models[1] and *causal Bayesian networks* (Pearl 2000). Both are typically described using directed acyclic graphs, where each node is associated with a variable. In a causal model, with each variable $Y$ associated with a non-root node, there is a deterministic (structural) equation, that gives the value of $Y$ as a function of the values of its parents; there is also a probability on the values of root nodes. In a CBN, like in a Bayesian network, each variable $Y$ is associated with a *conditional probability*

---

[1]Unless specified, in this paper, "causal models" refer to "functional causal models".

*table (cpt)*, that for each setting of the parents of $Y$, gives the probability of $Y$ conditional on that setting. In a functional causal model, it is actually straightforward to determine the conditional probability of formulas involving interventions. In a CBN, this is far from true. Indeed, recent work of Beckers (2023) has shown that an approach given by Pearl (2000) to calculate these probabilities in a CBN is incorrect. [2] Pearl also calculates probabilities in a CBN by implicitly reducing the CBN to a family of functional causal models (see, e.g., (Pearl 2000, Theorem 9.2.10)), but he does not give an explicit reduction, nor does he give a formal definition of the probability of a formula in a CBN. Here, we do both. Using this approach leads to formulas having a range of probabilities in a CBN, whereas in a functional causal model, their probability is unique.

But we take an additional significant step. Pearl assumes that mechanisms that determine how interventions work (which are given by the cpts in the case of CBNs and the structural equations in the case of causal models) are *autonomous*: as Pearl puts it, "external changes affecting one equation do not imply changes to the others" (Pearl 2000, p. 28). We model this autonomy formally by taking the equations to be independent of each other, in an appropriate space. As shown recently by Richardson, Peters, and Halpern (2024), taking the equations that characterize different variables to be independent is a necessary and sufficient condition for reproducing all the (conditional) independencies in the underlying Bayesian network, as determined by *d-separation* (Pearl 1988). Thus, this independence seems like a natural and critical assumption to get CBNs and causal models to work as we would expect.

Here we assume that, not only are the equations that define different variables independent, but also the equations that give the values of a variable for different settings of its parents. We never need to consider the values of a variable for different settings of its parents in a standard Bayesian network, but this is necessary to determine the probability of a formula involving interventions, such as $X = 0 \land Y = 0 \land [X \leftarrow 1](Y = 1)$ ($X$ and $Y$ have value 0, but if $X$ is set to 1, $Y$ gets value 1). Taking these latter equations to be independent is not always appropriate;[3] For example, there may be a latent exogenous variable that affects the value of $Y$ for different settings of $Y$'s parents. But if the parents of $Y$ (including exogenous variables) are all observable, and screen $Y$ off from the effects of all other variables, then the independence assumption seems appropriate.

Making these independence assumptions has significant benefits. For one thing, it allows us to uniquely identify the probability of queries in a CBN; rather than getting a range of values, we get a unique value. Moreover, for many formulas of interest (including the *probability of necessity* and *probability of sufficiency* (Pearl 2000), we can compute the probability by considering only conditional probabilities involving only a subset of endogenous and exogenous variables, which do not involve interventions. This means that these probabilities can be estimated from observational data, without requiring involving controlled experiments. This can have huge implications in settings where such experimental data is not available but the exogenous variables can be observed.

The rest of this paper is organized as follows. Section 2 reviews the formalism of causal models. Section 3 gives semantics to formulas in Causal Bayesian Networks (CBNs) and Section 4 shows that any CBN can be converted to a compatible causal model that satisfies the independence assumptions that we are interested in. We show how counterfactual probabilities of necessity and sufficiency can be simplified and calculated in the appendix.

---

[2] Pearl (2000)[Theorem 7.1.7] provides a (correct) three-step procedure for calculating counterfactual probabilities in a causal model. But then on p. 220, Pearl says that the same procedure works for CBNs. Specifically, he says "counterfactual probabilities $p(Y_x = y \mid e)$ can still be evaluated using the three steps (abduction, action, and prediction) of Theorem 7.1.7. In the abduction phase, we condition the prior probability $p(u)$ of the root nodes on the evidence available, $e$, and so obtain $p(u \mid e)$. In the action phase, we delete the arrows entering variables in set $X$ and instantiate their values to $X = x$. Finally, in the prediction phase, we compute the probability of $Y = y$ resulting from the updated manipulated network." As Beckers shows, this is incorrect. Here's a trivial counterexample. Suppose that we have a simple causal model with one exogenous variable $U$, which is the parent of an endogenous variable $Y$, which in turn in is the parent of an endogenous variable $X$. All variables are binary. $U = 1$ with probability 1. $Y = U$, and if $Y = 1$, then $X = 0$ with probability $1/2$ and $X = 1$ with probability $1/2$. Now consider $p(X = 1 \mid X = 1)$. Applying Pearl's procedure, the probability of $U = 1$ continues to be 1 (no amount of conditioning will change that). Since there are no interventions, $Y = 1$ with probability 1, and $X = 1$ with probability $1/2$. That is, $p(X = 1 \mid X = 1) = 1/2$ according to Pearl's procedure. But this is clearly incorrect. Beckers provides a more general counterexample, and shows that the problem is not fixable in any obvious way.

[3] We thank Elias Bareinboim and Scott Muller for stressing this point.

## 2 Causal Models and CBNs

In a (*functional*) *causal model* (also called a *structural equations model*), the world is assumed to be described in terms of variables and their values. Some variables may have a causal influence on others. This influence is modeled by a set of *structural equations*. It is conceptually useful to split the variables into two sets: the *exogenous* variables, whose values are determined by factors outside the model, and the *endogenous* variables, whose values are ultimately determined by the exogenous variables. In some settings, exogenous variables can be observed; but they can never be intervened upon, as (by assumption) their values are determined by factors outside the model. Note that exogenous variables may involve latent factors that are not observable, and may even be unknown. For example, in an agricultural setting, we could have endogenous variables that describe crop produce, amount of fertilizers used, water consumption, and so on, and exogenous variables that describe weather conditions (which cannot be modified, but can be observed) and some latent factors, like the activity level of pollinators (which cannot be observed or measured). The structural equations describe how the values of endogenous variables are determined (e.g., how the water consumption depends on the weather conditions and the amount of fertilizer used).

Formally, a *causal model* $M$ is a pair $(\mathcal{S}, \mathcal{F})$, where $\mathcal{S}$ is a *signature*, which explicitly lists the endogenous and exogenous variables and characterizes their possible values, and $\mathcal{F}$ defines a set of *modifiable structural equations*, relating the values of the variables. A signature $\mathcal{S}$ is a tuple $(\mathcal{U}, \mathcal{V}, \mathcal{R})$, where $\mathcal{U}$ is a set of exogenous variables, $\mathcal{V}$ is a set of endogenous variables, and $\mathcal{R}$ associates with every variable $Y \in \mathcal{U} \cup \mathcal{V}$ a nonempty set $\mathcal{R}(Y)$ of possible values for $Y$ (that is, the set of values over which $Y$ *ranges*). For simplicity, we assume that $\mathcal{V}$ is finite, as is $\mathcal{R}(Y)$ for every endogenous variable $Y \in \mathcal{V}$. $\mathcal{F}$ associates with each endogenous variable $X \in \mathcal{V}$ a function denoted $F_X$ such that $F_X : (\times_{U \in \mathcal{U}} \mathcal{R}(U)) \times (\times_{Y \in \mathcal{V} - \{X\}} \mathcal{R}(Y)) \rightarrow \mathcal{R}(X)$. This mathematical notation just makes precise the fact that $F_X$ determines the value of $X$, given the values of all the other variables in $\mathcal{U} \cup \mathcal{V}$.

The structural equations define what happens in the presence of external interventions. Setting the value of some variable $X$ to $x$ in a causal model $M = (\mathcal{S}, \mathcal{F})$ results in a new causal model, denoted $M_{X \leftarrow x}$, which is identical to $M$, except that the equation for $X$ in $\mathcal{F}$ is replaced by $X = x$.

Following most of the literature, we restrict attention here to what are called *recursive* (or *acyclic*) models. In such models, there is a total ordering $\prec$ of the endogenous variables such that if $X \prec Y$, then $X$ is not causally influenced by $Y$, that is, $F_X(\ldots, y, \ldots) = F_X(\ldots, y', \ldots)$ for all $y, y' \in \mathcal{R}(Y)$. If $X \prec Y$, then the value of $X$ may affect the value of $Y$, but the value of $Y$ cannot affect the value of $X$. It should be clear that if $M$ is an acyclic causal model, then given a *context*, that is, a setting $\vec{u}$ for the exogenous variables in $\mathcal{U}$, there is a unique solution for all the equations. We simply solve for the variables in the order given by $\prec$.

A recursive causal model can be described by a dag (directed acyclic graph) whose nodes are labeled by variables, and there is an edge from $X$ to $Y$ if $X \prec Y$. We can assume without loss of generality that the equation for $Y$ involves only the parents of $Y$ in the dag. The roots of the dag are labeled by exogenous variables or endogenous variables with no parents; all the remaining nodes are labeled by endogenous variables.[4]

A *probabilistic* (functional) causal model is a pair $(M, \Pr)$ consisting of a causal model $M$ and a probability $\Pr$ on the contexts of $M$. In the rest of this paper, when we refer to a "causal model", we mean a probabilistic functional causal model, unless we explicitly say otherwise.

A *causal Bayesian network (CBN)* is a tuple $M = (\mathcal{S}, \mathcal{P})$ described by a signature $\mathcal{S}$, just like a causal model, and a collection $\mathcal{P}$ of *conditional probability tables (cpts)*, one for each (endogenous and exogenous) variable.[5] For this paper, we focus on recursive CBNs that can be characterized by a dag, where there is a bijection between the nodes and the (exogenous and endogenous) variables. The cpt for a variable $X$ quantifies the effects of the parents of $X$ on $X$. For example, if the parents of $X$ are $Y$ and $Z$ and all variables are binary, then the cpt for $X$ would have entries for all $j, k \in \{0, 1\}^2$, where the entry for $(j, k)$ describes$\{\Pr(X = 0 \mid Y = j, Z = k)$. (There is no need to have

---

[4]Note that the equation for an endogenous variable $X$ with no parents must be a constant function; e.g., $F_X = 3$. In the model $M_{X \leftarrow x}$ that results from $M$ after intervening on $X$, $X$ is an endogenous variable with no parents.

[5]Some authors (e.g., Pearl (2000) seem to assume that CBNs do not include exogenous variables. We find it useful to allow them.

an explicit entry for $P(X = 1 \mid Y = j \cap Z = k)$, since this is just $1 - P(X = 0 \mid Y = j \cap Z = k)$.)
The cpt for a root of the dag is just an unconditional probability, since a root has no parents.

Just as for causal models, we can also perform interventions in a CBN: intervening to set the value of some variable $X$ to $x$ in a CBN $M$ results in a new CBN, denoted $M_{X \leftarrow x}$, which is identical to $M$, except that now $X$ has no parents; the cpt for $X$ just gives $X$ value $x$ with probability 1.

Note that we typically use the letter $M$ to refer to both non-probabilistic causal models and CBNs, while we use $\mathrm{Pr}$ to refer to the probability on contexts in a probabilistic causal model. We use $P$ to refer to the probability in a cpt. It is also worth noting that a causal model can be viewed as a CBN; the equation $Y = F(\vec{x})$ can be identified with the entry $P(Y = F(\vec{x})) \mid \vec{X} = \vec{x}) = 1$ in a cpt.

## 3   Giving semantics to formulas in CBNs

### 3.1   The problem

Consider the following (standard) language for reasoning about causality: Given a signature $\mathcal{S} = (\mathcal{U}, \mathcal{V}, \mathcal{R})$, a *primitive event* is a formula of the form $X = x$, for $X \in \mathcal{V}$ and $x \in \mathcal{R}(X)$. A *causal formula (over $\mathcal{S}$)* is one of the form $[Y_1 \leftarrow y_1, \ldots, Y_k \leftarrow y_k]\varphi$, where $\varphi$ is a Boolean combination of primitive events, $Y_1, \ldots, Y_k$ are distinct variables in $\mathcal{V}$, and $y_i \in \mathcal{R}(Y_i)$. Such a formula is abbreviated as $[\vec{Y} \leftarrow \vec{y}]\varphi$. The special case where $k = 0$ is abbreviated as $\varphi$. Intuitively, $[Y_1 \leftarrow y_1, \ldots, Y_k \leftarrow y_k]\varphi$ says that $\varphi$ would hold if $Y_i$ were set to $y_i$, for $i = 1, \ldots, k$. $\mathcal{L}(\mathcal{S})$ is the language consisting of Boolean combinations of causal formulas. We typically take the signature $\mathcal{S}$ to be fixed, and just write $\mathcal{L}$. It will be convenient to consider a slightly richer language, that we denote $\mathcal{L}^+(\mathcal{S})$. It extends $\mathcal{L}(\mathcal{S})$ by allowing primitive events $U = u$, where $U \in \mathcal{U}$, and also allowing interventions on exogenous variables.[6]

A pair $(M, \vec{u})$ consisting of a (non-probabilistic) causal model $M$ and a context $\vec{u}$ is called a *(causal) setting*. A formula $\varphi \in \mathcal{L}^+$ is either true or false in a setting. We write $(M, \vec{u}) \models \varphi$ if the causal formula $\varphi$ is true in the setting $(M, \vec{u})$. The $\models$ relation is defined inductively. $(M, \vec{u}) \models X = x$ if the variable $X$ has value $x$ in the unique (since we are dealing with acyclic models) solution to the equations in $M$ in context $\vec{u}$ (that is, the unique vector of values for the endogenous variables that simultaneously satisfies all equations in $M$ with the variables in $\mathcal{U}$ set to $\vec{u}$). The truth of conjunctions and negations is defined in the standard way. Finally, $(M, \vec{u}) \models [\vec{Y} \leftarrow \vec{y}]\varphi$ if $(M_{\vec{Y} \leftarrow \vec{y}}, \vec{u}_{\vec{Y} \leftarrow \vec{y}}) \models \varphi$, where $(M_{\vec{Y} \leftarrow \vec{y}}$ is identical to $M$ except that the equation for each endogenous variable $Y \in \vec{Y}$ is replaced by $Y = y^*$, where $y^* \in \mathcal{R}(Y)$ is the value in $\vec{y}$ corresponding to $Y$, and $\vec{u}_{\vec{Y} \leftarrow \vec{y}}$ is identical to $\vec{u}$, except that for each exogenous variable $U \in \vec{Y}$, the component of $\vec{u}$ corresponding to $U$ is replaced by $u^*$, where $u^* \in \mathcal{R}(U)$ is the value in $\vec{y}$ corresponding to $U$. (We remark that in a CBN, intervening to set an exogenous variable $U$ to $u^*$ is just like any other intervention; we change the cpt for $U$ so that $u^*$ gets probability 1.)

In a probabilistic causal model $(M, \mathrm{Pr})$, we can assign a probability to formulas in $\mathcal{L}$ by taking the probability of a formula $\varphi$ in $M$, denoted $\mathrm{Pr}(\varphi)$, to be $\mathrm{Pr}(\{\vec{u} : (M, \vec{u}) \models \varphi\})$. Thus, the probability of $\varphi$ in $M$ is simply the probability of the set of contexts in which $\varphi$ is true; we can view each formula as corresponding to an event.

When we move to CBNs, things are not so straightforward. First, while we still have a probability on contexts, each context determines a probability on *states*, assignments of values to variables. A state clearly determines a truth value for formulas that do not involve interventions; call such formulas *simple formulas*. Thus, we can compute the truth of a simple formula $\varphi$ in a context, and then using the probability of contexts, determine the probability of $\varphi$ in a CBN $M$. But what about a causal formula such as $\psi = [\vec{Y} \leftarrow \vec{y}]\varphi$? Given a context $\vec{u}$, we can determine the model $M' = M_{\vec{Y} \leftarrow \vec{y}}$. In $(M', \vec{u})$, $\varphi$ is an event whose probability we can compute, as discussed above. We can (and will) take this probability to be the probability of the formula $\psi$ in $(M, \vec{u})$. But note that $\psi$ does not correspond to an event in $M$, although we assign it a probability.

---

[6] It is conceptually somewhat inconsistent to allow interventions on exogenous variables, since their value is assumed to be determined by factors outside the model, but it is technically convenient for some of our results.

It gets harder to evaluate probability if we add another conjunct $\psi'$ and consider the formula $\psi \wedge \psi'$. While we can use the procedure above to compute the probability of $\psi$ and $\psi'$ individually in $(M, \vec{u})$, what is the probability of the conjunction? Because such formulas do not correspond to events in $M$, this is not obvious. We give one approach for defining the probability of a formula in a CBN by making one key assumption, which can be viewed as a generalization of Pearl's assumption. Pearl assumes that mechanisms that determine how interventions work (which are the cpts in the case of CBNs and the structural equations in the case of causal models) are *autonomous*; he takes that to mean "it is conceivable to change one such relationship without changing the others" (Pearl 2000, p. 22). We go further and assume, roughly speaking, that they are (probabilistically) independent. In a causal model, the mechanism for a given variable (specifically, the outcome after the intervention) is an event, so we can talk about mechanisms being independent. While it is not an event in a CBN, we nevertheless use the assumption that mechanisms are independent to guide how we determine the probability of formulas in $\mathcal{L}$ in a CBN.

### 3.2 Independence of cpts and complete combinations of conditional events

To describe our approach, we must first make clear what we mean by mechanisms (cpts) being independent. This has two components: the outcomes of cpts for different variables are independent, and for the cpt for a single variable Y, the outcomes for different settings of the parents of $Y$ are independent. Indeed, all these outcomes are mutually independent. We believe that these independence assumptions are quite reasonable and, capture the spirit of Bayesian networks. In fact, Richardson, Peters, and Halpern (2024) show that the assumption that cpts involving different variables are independent is equivalent to the (conditional) independence assumptions made in Bayesian networks (see Section 3.4 for further discussion).

In more detail, suppose that we have a variable $Y_1$ in a CBN $M$ with parents $X_1 \ldots, X_m$. We want to consider events of the form $Y_1 = y_1 \mid (X_1 = x_1, \ldots, X_m = x_m)$, which we read "$Y_1 = y_1$ given that $X_1 = x_1$, ..., and $X_m = x_m$". Such events have a probability, given by the cpts for $Y_1$. We call such an event a *conditional event for CBN $M$*. (Explicitly mentioning the CBN $M$ is necessary, since on the right-hand side of the conditional with left-hand side $Y$, we have all the parents of $Y$; what the parents are depends on $M$.) Roughly speaking, we identify such a conditional event with the formula $[X_1 \leftarrow j_1, \ldots, X_m \leftarrow j_m](Y_1 = 1)$. This identification already hints at why we we care about conditional events (and their independence). Suppose for simplicity that $m = 1$. To determine the probability of a formula such as $X_1 = 0 \wedge Y_1 = 0 \wedge [X_1 \leftarrow 1](Y_1 = 1)$ we need to apply both the entry in the cpt for $Y_1 = 0 \mid X_1 = 0$ and the entry for $Y_1 = 1 \mid X = 1$. They each give a probability; the probability of the formula $X_1 = 0 \wedge Y_1 = 0 \wedge [X_1 \leftarrow 1](Y_1 = 1)$ is the probability that the conditional events $Y_1 = 0 \mid X_1 = 0$ and $Y_1 = 1 \mid X = 1$ hold simultaneously. Our independence assumption implies that this probability is the product of the probability that each of them holds individually (which is given by the cpt for $Y_1$).

This is an instance of independence within a cpt; we want the conditional events in a cpt for a variable $Y$ for different settings of the parents of $Y$ to be independent. (Of course, conditional events for the same setting of the parents, such as $Y_1 = 0 \mid X_1 = 1$ and $Y_1 = 1 \mid X_1 = 1$, are not independent.) Independence for cpts of different variables is most easily explained by example: Suppose that $Y_2$ has parents $X_1$ and $X_3$. Then we want the events $Y_1 = 0 \mid X_1 = 0$ and $Y_2 = 1 \mid (X_1 = 0, X_3 = 1)$ to be independent. This independence assumption will be needed to compute the probability of formulas such as $[X_1 \leftarrow 0](Y_1 = 0) \wedge [X_1 = 0, X_3 = 1](Y_2 = 1)$. As we said, we in fact want to view all the relevant conditional events as *mutually* independent.[7]

Although we use the term "conditional event", these are not events in a CBN. On the other hand, in a causal model, there are corresponding notions that really do correspond to events. For example, the conditional event $Y_1 = 0 \mid X_1 = 1$ corresponds to the set of contexts where the formula $[X_1 \leftarrow 1](Y_1 = 1)$ is true. Starting with a CBN $M$, we will be interested in causal models for which the probability $P(Y_1 = 0 \mid X_1 = 1)$, as given by the cpt for $Y_1$ in $M$, is equal to the probability of the corresponding event in the causal model.

Going back to CBNs, define a *complete combination of conditional events (ccce) for $M$* to be a conjunction consisting of the choice of one conditional event for $M$ for each endogenous variable

---

[7]This implicitly assumes that all exogenous variables are independent. We can easily drop this assumption by assuming that rather than having a separate cpt for each exogenous variable, we just have a single cpt for contexts. Nothing in the rest of the discussion would change if we did this.

$X$ and each setting of the parents of $X$. A *fixed-context* ccce (fccce) involves fewer conjuncts; we have only conditional events where for all the exogenous parents $U$ of a variable $X$, the value of $U$ is the same as its value in the conjunct determining the value of $U$ (the examples should make clear what this means).

**Example 3.1.** *Consider the CBN $M^*$ with the following dag:*

*, where all variables are binary, and the cpts give the following probabilities: $P(U = 0) = a$, $P(X = 0 \mid U = 0) = b$, $P(X = 0 \mid U = 1) = c$, $P(Y = 0 \mid X = 0) = d$, and $P(Y = 0 \mid X = 1) = e$. Then a ccce consists of 5 conjuncts:*

- *one of $U = 0$ and $U = 1$;*

- *one of $X = 0 \mid U = 0$ and $X = 1 \mid U = 0$;*

- *one of $X = 0 \mid U = 1$ and $X = 1 \mid U = 1$;*

- *one of $Y = 0 \mid X = 0$ and $Y = 1 \mid X = 0$; and*

- *one of $Y = 0 \mid X = 1$ and $Y = 1 \mid X = 1$.*

*An fccce consist of only 4 conjuncts; it has only one of the second and third conjuncts of a ccce. In particular, if $U = 0$ is a conjunct of the fccce, then we have neither $X = 0 \mid U = 1$ nor $X = 1 \mid U = 1$ as a conjunct; similarly, if $U = 1$ is a conjunct, then we have neither $X = 0 \mid U = 0$ nor $X = 1 \mid U = 0$ as a conjunct. (This is what we meant above by saying that each exogenous parent $U$ of $X$ must have the same value as in conjunct that determines $U$'s value.)*

It is not hard to show that, in this case, there are 32 ccces and 16 fccces. Moreover, (in this example and in general) each fccce is equivalent to a disjunction of ccces. The number of ccces and fccces can be as high as doubly exponential (in the number of variables), each one involving exponentially many choices. For example, if a variable $Y$ has $n$ parents, each of them binary, there are $2^n$ possible settings of the parents of $Y$, and we must choose one value of $Y$ for each of these $2^n$ settings, already giving us $2^{2^n}$ choices. It is easy to see that there is also a double-exponential upper bound.

If we think of a conditional event of the form $Z = 1 \mid X = 0, Y = 0$ as saying "if $X$ were (set to) 0 and $Y$ were (set to) 0, then $Z$ would be 1", then given a ccce and a formula $\varphi \in \mathcal{L}$ and context $\vec{u}$, we can determine if $\varphi$ is true or false. We formalize this shortly. We can then take the probability of $\varphi$ to be the sum of the probabilities of the ccces that make $\varphi$ true. The probability of a ccce is determined by the corresponding entry of the cpt. Thus, if we further assume independence, we can determine the probability of each ccce, and hence the probability of any formula $\varphi$. We now give some informal examples of how this works, and then formalize the procedure in Section 3.3.

**Example 3.2.** *In the CBN $M^*$ described in Example 3.1, there are two fccces where $\varphi = X = 0 \wedge Y = 0 \wedge [X \leftarrow 1](Y = 1)$ is true: (a) $U = 0 \wedge (X = 0 \mid U = 0) \wedge (Y = 0 \mid X = 0) \wedge (Y = 1 \mid X = 1)$; and (b) $U = 1 \wedge (X = 0 \mid U = 1) \wedge (Y = 0 \mid X = 0) \wedge (Y = 1 \mid X = 1)$. Each of these two fccces is the disjunction of two ccces, which extend the fccce by adding a fifth conjunct. For example, for the first fccce, we can add either the conjunct $X = 0 \mid U = 1$ or the conjunct $X = 1 \mid U = 1$. The total probability of these two fccces is $abd(1 - e) + (1 - a)cd(1 - e)$; this is the probability of $\varphi$ in $M^*$.*

We give one more example of this calculation.

**Example 3.3.** *Consider the model CBN $M^\dagger$, which differs from $M^*$ in that now $U$ is also a parent of $Y$; the dag is shown below. $M^*$ and $M^\dagger$ have the same cpts for $U$ and $X$; the cpt of $Y$ in $M^\dagger$ is $P(Y = 0 \mid U = 0, X = 0) = f_1$, $P(Y = 0 \mid U = 0, X = 1) = f_2$, $P(Y = 0 \mid U = 1, X = 0) = f_3$, $P(Y = 0 \mid U = 1, X = 1) = f_4$.*

*Now there are 128 ccces, but only 16 fccces; the formula $\varphi = X = 0 \wedge Y = 0 \wedge [X \leftarrow 1](Y = 1)$ is true in only two of these fccces: (a) $U = 0 \wedge (X = 0 \mid U = 0) \wedge (Y = 0 \mid (U = 0, X = 0)) \wedge (Y = 1 \mid (U = 0, X = 1))$; and (b) $U = 1 \wedge X = 0 \mid U = 1) \wedge (Y = 0 \mid (U = 1, X = 0)) \wedge (Y = 1 \mid (U = 1, X = 1))$. It is easy to check that $\Pr_{M^\dagger}(\varphi) = abf_1(1 - f_2) + (1 - a)cf_3(1 - f_4)$. The calculation of the probability of $\varphi$ is essentially the same in $M^*$ and $M^\dagger$.*

We denote by $\Pr_M(\varphi)$ the probability of a formula $\varphi$ in a CBN or causal model $M$. (We provide a formal definition of $\Pr_M(\varphi)$ for a CBN $M$ at the end of Section 3.)

### 3.3 Giving semantics to formulas in CBNs

We already hinted in Examples 3.2 and 3.3 how we give semantics to formulas in CBNs. We now formalize this.

The first step is to show that a ccce (resp., fccce) determines the truth of a formula in $\mathcal{L}^+(\mathcal{S})$ (resp., $\mathcal{L}(\mathcal{S})$) in a causal model. To make this precise, we need a few definitions. We take the *type* of a CBN $M = (\mathcal{S}, \mathcal{P})$, where $\mathcal{S} = (\mathcal{U}, \mathcal{V}, \mathcal{R})$ to consist of its signature $\mathcal{S}$ and, for each endogenous variable, a list of its parents (which is essentially given by the dag associated with $M$, without the cpts). A causal model $M' = (\mathcal{S}', \mathcal{F}')$ has the same type as $M$ if $\mathcal{S}' = (\mathcal{U} \cup \mathcal{U}', \mathcal{V}, \mathcal{R}')$, where $\mathcal{U}'$ is arbitrary, $\mathcal{R}'|_{\mathcal{U} \cup \mathcal{V}} = \mathcal{R}$, and $\mathcal{F}'$ is such that each endogenous variable $X$ depends on the same variables in $\mathcal{U} \cup \mathcal{V}$ according to $\mathcal{F}'$ as it does according to the type of $M$ (but may also depend on any subset of $\mathcal{U}'$).

**Definition 3.4.** *For the conditional event $Y = y \mid (X_1 = x_1, \ldots, X_m = x_m)$, let the corresponding formula be $[X_1 \leftarrow x_1, \ldots, X_m \leftarrow x_m](Y = y)$. (Note that the corresponding formula may be in $\mathcal{L}^+ - \mathcal{L}$, since some of the $X_i$s may be exogenous.) Let $\varphi_\alpha \in \mathcal{L}^+(\mathcal{S})$, the formula corresponding to the ccce $\alpha$, be the conjunction of the formulas corresponding to the conditional events in $\alpha$. We can similarly define the formula corresponding to an fccce.*

**Example 3.5.** *In the model $M^\dagger$ of Example 3.3, if $\alpha$ is the fccce $U = 0 \wedge (X = 0 \mid U = 0) \wedge (Y = 0 \mid (U = 0, X = 0)) \wedge (Y = 1 \mid (U = 0, X = 1))$, then $\varphi_\alpha$ is $U = 0 \wedge [U \leftarrow 0]X = 0 \wedge [U \leftarrow 0, X \leftarrow 0](Y = 0) \wedge [U \leftarrow 0, X \leftarrow 1](Y = 1)$.*

Say that a formula $\psi$ is *valid with respect to a CBN $M$* if $(M', \vec{u}) \models \psi$ for all causal settings $(M', \vec{u})$, where $M'$ is a causal model with the same type as $M$. The following theorem makes precise the sense in which a ccce determines whether or not an arbitrary formula is true.

**Theorem 3.6.** *Given a CBN $M = (\mathcal{S}, \mathcal{P})$ and a ccce (resp., fccce) $\alpha$, then for all formulas $\psi \in \mathcal{L}^+(\mathcal{S})$ (resp., $\psi \in \mathcal{L}(\mathcal{S})$) either $\varphi_\alpha \Rightarrow \psi$ is valid with respect to $M$ or $\varphi_\alpha \Rightarrow \neg\psi$ is valid with respect to $M$.*

**Proof:** We show that if two causal models $M_1$ and $M_2$ have the same type as $M$ and $\vec{u}_1$ and $\vec{u}_2$ are contexts such that $(M_1, \vec{u}_1) \models \varphi_\alpha$ and $(M_2, \vec{u}_2) \models \varphi_\alpha$, then for all formulas $\psi \in \mathcal{L}^+(\mathcal{S})$ (resp., $\psi \in \mathcal{L}(\mathcal{S})$), we have that

$$(M_1, \vec{u}_1) \models \psi \text{ iff } (M_2, \vec{u}_2) \models \psi. \tag{1}$$

The claimed result follows immediately. The details of the proof can be found in the appendix. ∎

Based on this result, we can take the probability of a formula $\varphi \in \mathcal{L}^+(\mathcal{S})$ in a CBN $M$ to be the probability of the ccces that imply it. To make this precise, given a CBN $M$, say that a probabilistic causal model $(M', \Pr)$ is *compatible* with $M$ if $M'$ has the same type as $M$, and the probability $\Pr$ is such that all the cpts in $M$ get the right probability in $M'$. More precisely, for each endogenous variable $Y$ in $M$, if $X_1, \ldots, X_k$ are the parents of $Y$ in $M$, then for each entry $P(Y = y \mid X_1 = x_1, \ldots, X_k = x_k) = a$ in the cpt for $Y$, $\Pr$ is such that the corresponding formula $[X_1 \leftarrow x_1, \ldots, X_k \leftarrow x_k](Y = y)$ gets probability $a$. $(M', \Pr)$ is *i-compatible* with $M$ (the *i* stands for *independence*) if it is compatible with $M$ and, in addition, $\Pr$ is such that the events described by the formulas corresponding to entries for cpts for different variable (i.e. the set of contexts in $M$ that make these formulas true) are independent, as are the events described by the formulas corresponding to different entries for the cpt for a given variable. Thus, for example, if $(x_1', \ldots, x_k') \neq (x_1, \ldots, x_k)$, then we want the events described by $[X_1 \leftarrow x_1, \ldots, X_k \leftarrow x_k](Y = y)$ and $[X_1 \leftarrow x_1', \ldots, X_k \leftarrow x_k'](Y = y)$ to be independent (these are different entries of the cpt for $Y$); and if $Y' \neq Y$ and has parents $X_1', \ldots, X_m'$ in $M$, then we want the events described by $[X_1 \leftarrow x_1, \ldots, X_k \leftarrow x_k](Y = y)$ and $[X_1' \leftarrow x_1', \ldots, X_m \leftarrow x_m'](Y' = y')$ to be independent (these are entries of cpts for different variables).

**Theorem 3.7.** *Given a CBN $M$ and a formula $\varphi \in \mathcal{L}^+(\mathcal{S})$, the probability of $\varphi$ is the same in all causal models $M'$ i-compatible with $M$.*

**Proof:** It follows from Theorem 3.6 that the probability of $\varphi$ is the sum of the probabilities of the formulas $\varphi_\alpha$ for the ccces $\alpha$ such that $\varphi_\alpha \Rightarrow \varphi$ is valid. It is immediate that these formulas have the same probability in all causal models i-compatible with $M$. ∎

Formally, we take $\Pr_M(\varphi)$, the probability of $\varphi$ in the CBN $M$, to be $\Pr_{M'}(\varphi)$ for a causal model $M'$ i-compatible with $M$. By Theorem 3.7, it does not matter which causal model $M'$ i-compatible with $M$ we consider. Note for future reference that if we consider only causal models compatible with $M$, dropping the independence assumption, we would get a range of probabilities.

### 3.4 Discussion

Four points are worth making: First, note that this way of assigning probabilities in a CBN $M$ always results in the probability of a formula $\varphi \in \mathcal{L}^+$ being a sum of products of entries in the cpt. Thus, we can in principle compute the probabilities of (conditional) events involving interventions from observations of statistical frequencies (at least, as long as all settings of the parents of a variable in the relevant entries of the cpt have positive probability).

Second, the number of ccces may make the computation of the probability of a formula in a CBN seem unacceptably high. As the examples above shows, in practice, it is not so bad. For example, we typically do not actually have to deal with ccces. For one thing, it follows from Theorem 3.6 that to compute the probability of $\varphi \in \mathcal{L}$, it suffices to consider fccces. Moreover, when computing $\Pr_M(\varphi)$ where $\varphi$ involves an intervention of the form $X \leftarrow x$, we can ignore the entries in the cpts involving $X$, and for variables for which $X$ is a parent, we consider only entries in the cpts where $X = x$. We can also take advantage of the structure of the formula whose probability we are interested in computing to further simplify the computation, although the details are beyond the scope of this paper.

Third, as mentioned above, a formula involving interventions does not correspond in an obvious way to an event in a CBN, but it does correspond to an event in a (functional) causal model. The key point is that in a causal model, a context not only determines a state; it determines a state for every intervention. We can view a formula involving interventions as an event in a space whose elements are functions from interventions to worlds. Since a context can be viewed this way, we can view a formula involving interventions as an event in such a space. This makes conditioning on arbitrary formulas in $\mathcal{L}^+$ (with positive probability) in causal models well defined. By way of contrast, in a CBN, we can view a context as a function from interventions to distributions over worlds. Finally, it is worth asking how reasonable is the assumption that cpts are independent, that is, considering i-compatible causal models rather than just compatible causal models, which is what seems to have been done elsewhere in the literature (see, e.g., (Balke and Pearl 1994; Tian and Pearl 2000)).

As we said, Richardson, Peters, and Halpern (2024) show that the assumption that cpts involving different variables are independent is equivalent to the (conditional) independence assumptions made in Bayesian networks. More precisely, given a CBN $M$, let $M'$ be the non-probabilistic causal model constructed above. They show that if the probability $\Pr'$ makes interventions on different variables independent (i.e., if $\Pr'(\vec{U}, f_1, \ldots, f_m) = \Pr(\vec{u}) \times \Pr_{Y_1}(f_1) \times \cdots \times \Pr_{Y_m}(f_m)$, as in our construction), then all the conditional independencies implied by d-separation hold in $(M, \Pr')$ (see (Pearl 1988) for the formal definition of d-separation and further discussion). Conversely, if all the dependencies implied by d-separation hold in $(M, \Pr')$, then $\Pr'$ must make interventions on different variables independent.

This result says nothing about making interventions for different settings of the parents of a single variable independent. This is relevant only if we are interested in computing the probability of formulas such as $X = 0 \wedge Y = 0 \wedge [X \leftarrow 1](Y = 1)$, for which we need to consider (simultaneously) the cpt for $Y$ when $X = 0$ and when $X = 1$. As discussed earlier, independence is reasonable in this case if we can observe all the parents of a variable $Y$, and thus screen off $Y$ from the effects of all other variables (and other settings of the parents). We cannot always assume this, but in many realistic circumstances, we can. We give two general classes of examples where we can:

1. When debugging systems (including ml pipelines, database engines, or any general software) and network failures, users have access to all parameters related to the code and the execution environment (Fariha, Nath, and Meliou 2020; Kobayashi, Otomo, and Fukuda 2019; Galhotra, Fariha, Lourenço, Freire, Meliou, and Srivastava 2022). With this infor-

mation, a causal graph over different blocks of code, its parameters, and other environment variables like information about background processes can be constructed. This means we can address queries like "Given that component A is faulty, with what probability would repairing component B solve the problem" using our techniques.

2. Manufacturing pipelines across various industries, such as semiconductor fabrication, pharmaceuticals, automobile assembly, and battery production, typically consist of a series of interconnected stages. Each of these stages is equipped with several sensors designed to monitor and measure critical environmental conditions that directly impact the production process. These sensors collect data about variables such as temperature, humidity, pressure, and other factors that can influence the quality, efficiency, and consistency of the final product. For instance, in semiconductor fabrication, precise control of environmental conditions like temperature and humidity is crucial to ensuring the integrity of the microchips produced (Wu 2008). Similarly, in pharmaceutical manufacturing, sensors monitor parameters like pH levels and chemical concentrations to maintain the efficacy of the drugs being produced. Thus, we can answer queries like "what is the probability that a temperature increase of 3 degrees Celsius would result in a poor quality product, given that the humidity is high?".

Other potential applications of our framework include (a) modeling player performance in sports by considering factors like injury, skill, and sports facilities, (b) urban planning scenarios to analyze the impact of zoning laws, interest rates, and other factors on house prices, and (c) modeling agriculture yield by considering variables like soil quality and weather conditions.

## 4 Converting a CBN to a (Probabilistic) Causal Model

Our semantics for formulas in CBNs reduced to considering their semantics in i-compatible causal models. It would be useful to show explicitly that such i-compatible causal models exist and how to construct them. That is the goal of this section. Balke and Pearl (1994) sketched how this could be done. We largely follow and formalize their construction.

Starting with a CBN $M$, we want to construct an i-compatible probabilistic causal model $(M', \Pr')$, where $M'$ has the same type as $M$. To do this, for each endogenous variable $Y$ in $M$ with parents $X_1, \ldots, X_n$, we add a new exogenous variable $U_Y$; $\mathcal{R}(\mathcal{U}_Y)$ consists of all functions from $\mathcal{R}(X_1) \times \cdots \times \mathcal{R}(X_n)$ to $\mathcal{R}(Y)$. Balke and Pearl (1994) call such an exogenous variable a *response function*. (Response functions, in turn, are closely related to the *potential response variables* introduced by Rubin (1974).) We take $U_Y$ to be a parent of $Y$ (in addition to $X_1, \ldots, X_n$). We replace the cpt for $Y$ be the following equation for $Y$; $F_Y(x_1, \ldots, x_n, f) = f(x_1, \ldots, x_n)$, where $f$ is the value of $U_Y$. Since $f$ is a function from $\mathcal{R}(X_1) \times \cdots \times \mathcal{R}(X_n)$ to $\mathcal{R}(Y)$, this indeed gives a value of $Y$, as desired. Let $Y_1, \ldots, Y_m$ be the endogenous variables in $M$. We define the probability $\Pr'$ on $\mathcal{R}(\mathcal{U}) \times \mathcal{R}(U_{Y_1}) \times \cdots \times \mathcal{R}(U_{Y_m})$ by taking $\Pr'(\vec{u}, f_1, \ldots f_m) = \Pr(\vec{u}) \times \Pi_{i=1,\ldots,m} \Pr_{Y_i}(f_i)$, where $\Pr_{Y_i}$ reproduces the probability of the cpt for $Y_i$. Specifically, for an endogenous variable $Y$ with parents $X_1, \ldots, X_n$, $\Pr_Y(f) = \Pi_{\vec{x} \in \mathcal{R}(X_1) \times \cdots \times \mathcal{R}(X_n)} \Pr(Y = f(x_1, \ldots, x_n) \mid X_1 = x_1, \ldots, X_n = x_n)$. This makes interventions for different settings of $X_1, \ldots, X_n$ independent, which is essentially what we assumed in the previous section when defining the probability of formulas in $\mathcal{L}$ in $M_0$, in addition to making interventions on different variables independent and independent of the context in $M$. In any case, it is easy to see that this gives a well-defined probability on $\mathcal{R}(\mathcal{U}) \times \mathcal{R}(U_{Y_1}) \times \mathcal{R}(U_{Y_m})$, the contexts in $M'$. Moreover, $M'$ is clearly a causal model with the same type as $M$ that is i-compatible with $M$.

We can easily modify this construction to get a family of causal models compatible with $M$, by loosening the requirements on $\Pr'$. While we do want the marginal of $\Pr'$ on $\mathcal{U}$ to agree with the marginal of $\Pr$ on $\mathcal{U}$, and we want it to reproduce the probability of the cpt for each variable $Y_i$ (as defined above), there are no further independence requirements. If we do that, we get the bounds computed by Balke and Pearl (1994). The following example illustrates the impact of dropping the independence assumptions.

**Example 4.1.** *Consider the CBN $M^*$ from Example 3.1 again. Using the notation from that example, suppose that $a = 1$ and $b = d = 1/2$. Independence guarantees that the set of ccces that includes $U = 0$, $X = 0 \mid U = 0$, and $Y = 0 \mid X = 0$ has probability $abd = 1/4$. But now consider a causal model $(M^{**}, \Pr^{**})$ compatible with $M^*$ where the contexts are the*

*same as in our construction, but the probability* $\mathrm{Pr}^{**}$ *does not build in the independence assumptions of our construction. Recall that contexts in $M^{**}$ have the form $(u, f_X, f_Y)$. Since we want $(M^{**}, \mathrm{Pr}^{**})$ to be compatible with $M^*$, we must have $\mathrm{Pr}^{**}(\{(u, f_X, f_U, f_Y) : u = 0\}) = 1$, $\mathrm{Pr}^{**}(\{(u, f_X, f_Y) : f_X(0) = 0\}) = 1/2$, and $\mathrm{Pr}^{**}(\{(u, f_X, f_Y) : f_Y(0) = 0\}) = 1/2$, so that $\mathrm{Pr}^{**}$ agrees with the three cpts. But this still leaves a lot of flexibility. For example, we might have $\mathrm{Pr}^{**}(\{(u, f_X, f_Y) : f_X(0) = f_Y(0) = 0\} = Pr^{**}(\{(u, f_X, f_Y) : f_X(1) = f_Y(1) = 1\} = 1/2$ (so that $\mathrm{Pr}^{**}(\{(u, f_X, f_Y) : f_X(0) = 0, f_Y(1) = 1\}) = \mathrm{Pr}^{**}(\{(u, f_X, f_Y) : f_X(0) = 1, f_Y(1) = 0\}) = 0$). As shown in Example 3.2, $\mathrm{Pr}_{M^*}(X = 0 \wedge Y = 0 \wedge [X \leftarrow 1](Y = 1)) = 1/4$. However, it is easy to check that $\mathrm{Pr}_{M^{**}}(X = 0 \wedge Y = 0 \wedge [X \leftarrow 1](Y = 1)) = 1/2$. (Tian and Pearl (2000) give bounds on the range of probabilities for this formula, which is called the* probability of necessity*; see also Section B and (Pearl 2000, Section 9.2).)*

# 5 Computing counterfactual probabilities

In this section, we analyze *counterfactual probabilities*, introduced by Balke and Pearl (1994). Counterfactual probabilities have been widely used in several domains, including psychology (Hoerl, McCormack, and Beck 2011), epidemiology (Greenland and Robins 1999), and political science (Grynaviski 2013), to explain the effects on the outcome. More recently, they have proved useful in machine learning to explain the output of ML models (Beckers 2022).

Two types of counterfactual formulas that have proved particularly useful are the *probability of necessity* and the *probability of sufficiency*; we focus on them in this section. As discussed by Pearl (2000), counterfactual analysis is particularly useful when it comes to understanding the impact of a decision on the outcome. For example, we might be interested in the probability that an outcome $O$ would not have been favorable if $A$ were not true. This captures the extent to which $A$ is a *necessary* cause of $O$. Similarly, we might be interested in whether $A$ is *sufficient* for $O$: that is if $A$ were true, would $O$ necessarily be true? We now review the formal definitions of these notions; see (Pearl 2000) for more discussion.

**Definition 5.1.** *Let $X$ and $Y$ be binary variables in a causal model or CBN $M$.*

1. Probability of necessity of $X$ for $Y$: $\mathrm{PN}_M^{X,Y} = \mathrm{Pr}_M([X \leftarrow 0](Y = 0) | X = 1 \wedge Y = 1)$.

2. Probability of sufficiency of $X$ for $Y$: $\mathrm{PS}_M^{X,Y} = \mathrm{Pr}_M([X \leftarrow 1](Y = 1) \mid X = 0 \wedge Y = 0)$.

3. Probability of necessity and sufficiency of $X$ for $Y$: $\mathrm{PNS}_M^{X,Y} = \mathrm{Pr}_M([X \leftarrow 1](Y = 1) \wedge [X \leftarrow 0](Y = 0))$.

Pearl (2000) gives examples showing that neither the probability of necessity nor the probability of sufficiency in a CBN can be identified; we can just determine a range for these probabilities. But with our independence assumptions, they can be identified, justifying our notation. Moreover, these probabilities can be computed using only conditional probabilities of (singly) exponentially many simple formulas (not involving interventions). Since these formulas do not involve interventions, they can be estimated from observational data, without requiring involving controlled experiments. Thus, our results and assumptions have significant practical implications.

Let $Pa^X(Y)$ consist of all the parents of $Y$ other than $X$. For a set $\mathcal{Z}$ of variables, let $\mathcal{T}_{\mathcal{Z}}$ consist of all possible settings of the variables in $\mathcal{Z}$.

**Theorem 5.2.** *If $M$ is a CBN where $Y$ is a child of $X$, then*

(a) $\mathrm{PN}_M^{X,Y} = \sum_{c_{Pa^X(Y)}^j \in \mathcal{T}_{Pa^X(Y)}} \mathrm{Pr}_M(Pa^X(Y) = c_{Pa^X(Y)}^j \mid Y = 1 \wedge X = 1)$
$$\mathrm{Pr}_M(Y = 0 \mid X = 0 \wedge Pa^X(Y) = c_{Pa^X(Y)}^j);$$

(b) $\mathrm{PS}_M^{X,Y} = \sum_{c_{Pa^X(Y)}^j \in \mathcal{T}_{Pa^X(Y)}} \mathrm{Pr}_M(Pa^X(Y) = c_{Pa^X(Y)}^j \mid Y = 0 \wedge X = 0)$
$$\mathrm{Pr}_M(Y = 1 \mid X = 1 \wedge Pa^X(Y) = c_{Pa^X(Y)}^j);$$

(c) $\mathrm{PNS}_M^{X,Y} = \mathrm{PS}_M^{X,Y} \cdot \mathrm{Pr}_M(X = 0 \wedge Y = 0) + \mathrm{PN}_M^{X,Y} \cdot \mathrm{Pr}_M(X = 1 \wedge Y = 1)$.

We defer the proof of the theorem to Section B.1 in the appendix, where further extensions are also provided.

**Acknowledgments:** Halpern's work was supported in part by AFOSR grant FA23862114029, MURI grant W911NF-19-1-0217, ARO grant W911NF-22-1-0061, and NSF grant FMitF-2319186. Galhotra's work is supported by a grant from Infosys.

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

# A  Proof of Theorem 3.6

**Proof:** As we said, we show that if two causal models $M_1$ and $M_2$ have the same type as $M$ and $\vec{u}_1$ and $\vec{u}_2$ are contexts such that $(M_1, \vec{u}_1) \models \varphi_\alpha$ and $(M_2, \vec{u}_2) \models \varphi_\alpha$, then for all formulas $\psi \in \mathcal{L}^+(\mathcal{S})$ (resp., $\psi \in \mathcal{L}(\mathcal{S})$), we have that (1) (as defined in the main text) holds. The claimed result follows immediately.

We give the proof in the case that $\alpha$ is a ccce and $\psi \in \mathcal{L}^+(\mathcal{S})$. The modifications needed to deal with the case that $\alpha$ is an fccce and $\psi \in \mathcal{L}(\mathcal{S})$ are straightforward and left to the reader. Since $M$ is acyclic, we can order the exogenous and endogenous variables topologically. Let $X_1, \ldots, X_m$ be such an ordering. We first prove by induction on $j$ that, for all interventions $\vec{Y} \leftarrow \vec{y}$ (including the empty intervention) and $x_j \in \mathcal{R}(X_j)$, $(M_1, \vec{u}_1) \models [\vec{Y} \leftarrow \vec{y}](X_j = x_j)$ iff $(M_2, \vec{u}_2) \models [\vec{Y} \leftarrow \vec{y}](X_j = x_j)$.

For the base case, $X_1$ must be exogenous, and hence have no parents. If $X_1$ is not one of the variables in $\vec{Y}$, then we must have $(M_1, \vec{u}_1) \models [\vec{Y} \leftarrow \vec{y}](X_1 = x_1)$ iff $(M_1, \vec{u}_1) \models (X_1 = x_1)$, and similarly for $M_2$; since no variable in $\vec{Y}$ is a parent of $X_1$, intervening on $\vec{Y}$ has no effect on $X_1$. Since $(M_1, \vec{u}_1) \models \varphi_\alpha$ and $(M_2, \vec{u}_2) \models \varphi_\alpha$, $M_1$ and $M_2$ agree on the values of variables in $\mathcal{U}$. Thus, $(M_1, \vec{u}_1) \models (X_1 = x_1)$ iff $(M_2, \vec{u}_2) \models (X_1 = x_1)$. It follows that $(M_2, \vec{u}_2) \models [\vec{Y} \leftarrow \vec{y}](X_1 = x_1)$. $(M_2, \vec{u}_2) \models [\vec{Y} \leftarrow \vec{y}](X_1 = x_1)$, as desired.

On the other hand, if $X_1$ is one of the variables in $\vec{Y}$ (which can happen only if the formula is in $\mathcal{L}^+(\mathcal{S})$), let $x^*$ be the value in $\vec{y}$ corresponding to $X_1$. In that case, the formula $[\vec{Y} \leftarrow \vec{y}](X_1 = x^*)$ is valid with respect to $M$. It follows that $(M_1, \vec{u}_1) \models [\vec{Y} \leftarrow \vec{y}](X_1 = x_1)$ iff $x_1 = x^*$, and similarly for $(M_2, \vec{u}_2)$. The desired result follows. This completes the proof for the base case.

Now suppose that we have proved the result for $j < m$. Let $Z_1, \ldots, Z_k$ be the parents of $X_{j+1}$ in $M$. Since $X_1, \ldots, X_m$ is a topological sort, we must have $\{Z_1, \ldots, Z_k\} \subseteq \{X_1, \ldots, X_j\}$. Let $z_1, \ldots, z_k$ be values in $\mathcal{R}(Z_1), \ldots, \mathcal{R}(Z_k)$, respectively, such that $(M_1, \vec{u}_1) \models [\vec{Y} \leftarrow \vec{y}](Z_h = z_h)$, for $h = 1, \ldots, k$. By the induction hypothesis, $(M_2, \vec{u}_2) \models [\vec{Y} \leftarrow \vec{y}](Z_h = z_h)$, for $h = 1, \ldots, k$. Moreover, it is easy to see that $([\vec{Y} \leftarrow \vec{y}]\varphi \wedge [\vec{Y} \leftarrow \vec{y}]\varphi')) \Leftrightarrow [\vec{Y} \leftarrow \vec{y}](\varphi \wedge \varphi')$ is valid with respect to $M$. Thus, $(M_1, \vec{u}_1) \models [\vec{Y} \leftarrow \vec{y}](Z_1 = z_1 \wedge \ldots Z_k = z_k)$ and similarly for $(M_2, \vec{u}_2)$. Moreover, since $Z_1, \ldots, Z_k$ are the parents of $X_{j+1}$, it follows that $(M_1, \vec{u}_1) \models [\vec{Y} \leftarrow \vec{y}](X_{j+1} = x_{j+1})$ iff $[Z_1 = z_1 \wedge \ldots Z_k = z_k)](X_{j+1} = x_{j+1})$ is a conjunct of $\varphi_\alpha$. Since $(M_1, \vec{u}_1) \models \varphi_\alpha$ and $(M_2, \vec{u}_2) \models \varphi_\alpha$, the desired result follows, completing the induction proof.

The argument that $(M_1, \vec{u}_1) \models [\vec{Y} \leftarrow \vec{y}]\psi$ iff $(M_2, \vec{u}_2) \models [\vec{Y} \leftarrow \vec{y}]\psi$ for arbitrary (simple) formulas $\psi$ now follows from the fact that (as we already observed) $([\vec{Y} \leftarrow \vec{y}]\varphi \wedge [\vec{Y} \leftarrow \vec{y}]\varphi')) \Leftrightarrow [\vec{Y} \leftarrow \vec{y}](\varphi \wedge \varphi')$ is valid with respect to $M$, as are $([\vec{Y} \leftarrow \vec{y}]\varphi \vee [\vec{Y} \leftarrow \vec{y}]\varphi')) \Leftrightarrow [\vec{Y} \leftarrow \vec{y}](\varphi \vee \varphi')$ and $[\vec{Y} \leftarrow \vec{y}]\neg\varphi \Leftrightarrow \neg[\vec{Y} \leftarrow \vec{y}]\varphi$.

Finally, we can deal with Boolean combinations of causal formulas by a straightforward induction. This completes the argument that (1) holds for all formulas in $\psi \in \mathcal{L}^+(\mathcal{S})$. ∎

# B  Computing counterfactual probabilities (Missing Proof and Extension)

## B.1  Proof of Theorem 5.2

We prove the calculation for the probability of sufficiency, $\text{PS}_M^{X,Y}$. Essentially the same argument can be used to compute the probability of necessity, $\text{PN}_M^{X,Y}$. Finally, for part (c), we use the representation of $\text{PNS}_M^{X,Y}$ in terms of $\text{PS}_M^{X,Y}$ and $\text{PN}_M^{X,Y}$ given in (Pearl 2000, Lemma 9.2.6).

*Proof of Theorem 5.2 (b).* Let $\mathcal{Z} = \mathcal{U} \cup \mathcal{V} \setminus \{X, Y\}$. $\mathcal{T}_{\mathcal{Z}}$ has $2^{n-2}$ settings, where $n = |\mathcal{U} \cup \mathcal{V}|$. For a setting $c \in \mathcal{T}_{\mathcal{Z}}$, let $c_Z$ be the setting of the variable $Z$ in $c$.

By definition,

$$\text{PS}_M^{X,Y} = \frac{\Pr_M((X=0) \wedge (Y=0) \wedge [X \leftarrow 1](Y=1))}{\Pr_M((X=0) \wedge (Y=0))}. \tag{2}$$

Let the numerator $(X=0) \wedge (Y=0) \wedge [X \leftarrow 1](Y=1)$ be $\psi$. Then we have

$$\Pr_M(\psi) = \sum_{c \in \mathcal{T}_{\mathcal{Z}}} \Pr_M((X=0) \wedge (Y=0) \wedge [X \leftarrow 1](Y=1) \wedge \bigwedge_{Z \in \mathcal{Z}}(Z = c_Z)).$$

We next compute the probability of

$$\psi_c = (X=0) \wedge (Y=0) \wedge [X \leftarrow 1](Y=1) \wedge \bigwedge_{Z \in \mathcal{Z}}(Z = c_Z).$$

From Theorem 3.6, it follows that in all causal models $M'$ compatible with $M$,

$$\Pr_{M'}(\psi_c) = \sum_{\varphi_\alpha \implies \psi_c} \Pr_{M'}(\varphi_\alpha).$$

Now $\varphi_\alpha \implies \psi_c$ holds whenever $\alpha$ contains the following events:

1. $(X = 0 \mid Pa(X) = c_{Pa(X)})$

2. $(Y = 0 \mid X = 0, Pa^X(Y) = c_{Pa^X(Y)})$

3. $(Y = 1 \mid X = 1, Pa^X(Y) = c_{Pa^X(Y)})$

4. $(Z = c_Z \mid Pa(Z) = c_{Pa(Z)})$, for all $Z \in \mathcal{Z}$.

Let $S_c$ consist of all ccces that contain these four events, and let $\varphi_{S_c}$ be the conjunction of the formulas corresponding to the events in $S_c$. Then by Theorem 3.6,

$$\varphi_c \Leftrightarrow \bigvee_{\alpha \in S_c} \varphi_\alpha.$$

Since the formulas $\varphi_\alpha$ for distinct ccces in $S_c$ are mutually exclusive, we have that $\Pr_{M'}(\psi_c) = \Pr_{M'}(\varphi_S)$. Therefore,

$$
\begin{aligned}
&\Pr_{M'}(\psi) \\
&= \sum_{c \in \mathcal{T}_{\mathcal{Z}}} \Pr_{M'}(\varphi_c) \\
&= \sum_{c \in \mathcal{T}_{\mathcal{Z}}} \Pr_{M'}(X = 0 \wedge Y = 0 \wedge \bigwedge_{Z \in \mathcal{Z}}(Z = c_Z) \wedge [X \leftarrow 1, Pa^X(Y) \leftarrow c_{Pa^X(Y)}](Y = 1)) \\
&= \sum_{c_{Pa^X(Y)} \in \mathcal{T}_{Pa^X(Y)}} \Pr_{M'}(X = 0 \wedge Y = 0 \wedge Pa^X(Y) \leftarrow c_{Pa^X(Y)} \wedge [X \leftarrow 1, Pa^X(Y) \leftarrow c_{Pa^X(Y)}](Y = 1)).
\end{aligned}
$$

If $M'$ is i-compatible with $M$, then we can further conclude that

$$
\begin{aligned}
&\sum_{c \in \mathcal{T}_{\mathcal{Z}}} \Pr_{M'}(X = 0 \wedge Y = 0 \wedge \bigwedge_{Z \in \mathcal{Z}}(Z = c_Z) \wedge [X \leftarrow 1, Pa^X(Y) \leftarrow c_{Pa^X(Y)}](Y = 1)) \\
&= \sum_{c_{Pa^X(Y)} \in \mathcal{T}_{Pa^X(Y)}} \Pr_{M'}(X = 0 \wedge Pa^X(Y) = c_{Pa^X(Y)}) \Pr_{M'}(Y = 0 \mid X = 0 \wedge Pa^X(Y) = c_{Pa^X(Y)}) \\
&\qquad\qquad\qquad \Pr_{M'}(Y = 1 \mid X = 1 \wedge Pa^X(Y) = c_{Pa^X(Y)}) \\
&= \sum_{c_{Pa^X(Y)} \in \mathcal{T}_{Pa^X(Y)}} \Pr_{M'}(Y = 0 \wedge X = 0 \wedge Pa^X(Y) = c_{Pa^X(Y)}) \Pr_{M'}[Y = 1 \mid X = 1 \wedge Pa^X(Y) = c_{Pa^X(Y)}].
\end{aligned}
$$

Since $\Pr_{M'}(\psi) = \Pr_M(\psi)$, substituting the expression for $\Pr_M(\psi)$ into (2), we get

$$\text{PS}_M = \frac{\sum_{c_{Pa^X(Y)} \in \mathcal{T}_{Pa^X(Y)}} \Pr(Y = 0 \wedge X = 0 \wedge Pa^X(Y) = c_{Pa^X(Y)}) \Pr(Y = 1 \mid X = 1 \wedge Pa^X(Y) = c_{Pa^X(Y)})}{\Pr(X = 0 \wedge Y = 0)},$$

as desired. $\qquad \square$

We can extend Theorem 5.2 to the case where $Y$ is any descendant of $X$ (not necessarily a child of $X$). In this case, the term involving $Pa(Y)$ would change to the set of the ancestors of $Y$ at the same level as $X$ in the topological ordering of the variables. We can further extend Theorem 5.2 to arbitrary formulas $\psi$, where $\Pr(\psi)$ can be determined by calculating the probability of formulas that do not involve interventions (although they may involve conditional probabilities), and thus can be determined using only observational information. The key idea of the proof is to convert $\psi$ to a disjunction of conjunctions, where the disjuncts are mutually exclusive and have the form $\psi_i = \psi_{i0} \wedge \left( \bigwedge_{j \in \{1,\ldots,r\}} \psi_{ij} \right)$, where $\psi_{i0} = \left( \bigwedge_{j \in \{1,\ldots,s\}} (Z_{ij} = z_{ij}) \right)$ is a simple formula (with no intervention), and $\psi_{ij}$ for $j > 0$ has the form $[\vec{X}_j \leftarrow \vec{x}_j](\bigwedge_{k \in \{1,\ldots,t\}} Y_{ijk} = y_{ijk})$, where $Y_{ijk}$ is a descendant of $\vec{X}_j$ in $M$, so that we can apply the ideas in the proof of Theorem 5.2 to each disjunct separately. In terms of complexity, we show that $\Pr(\psi)$ can be estimated in $O(m \cdot 2^{nr^*})$ conditional probability calculations, where $r^*$ is the maximum number of conjuncts in a disjunction $\psi_i$ that involve at least one intervention, and $m$ is the number of disjuncts in the DNF. Unfortunately, for an arbitrary formula $\psi$, determining $\Pr(\psi)$ may involve doubly-exponentially many conditional probabilities. We defer details to Section **??** in the appendix.

**Theorem B.1.** *Given a CBN $M = (\mathcal{S}, \mathcal{P})$ and an arbitrary formila $\psi$, then $\Pr(\psi)$ can be determined by taking the probability of formulas that do not involve interventions (although they may involve conditional probabilities), and thus can be determined using only observational information.*

We now show that we can calculate the probability of an arbitrary formula $\psi$ in terms of conditional probabilities that can be estimated from observational data. To prove this result, we first convert $\psi$ to an equivalent formula in a canonical form. Specifically, it has the form $\psi_1 \vee \cdots \vee \psi_m$, where the $\psi_i$s are mutually exclusive and each $\psi_i$ is a conjunction of the form $\psi_{i0} \wedge \cdots \wedge \psi_{ir_i}$, where $\psi_{i0}$ is a simple formula and for $1 \leq j \leq r_i$, $\psi_{ij}$ is a formula of the form $[\vec{X}_j \leftarrow \vec{x}_j](\bigwedge_{k \in \{1,\ldots,t_{ij}\}} Y_{ijk} = y_{ijk})$, and the interventions are all distinct. This conversion just involves standard propositional reasoning and two properties which hold under the semantics described in Section 3. The first is that $[Y \leftarrow y]\varphi \wedge [Y \leftarrow y]\varphi'$ is equivalent to $[Y \leftarrow y](\varphi \wedge \varphi')$. The second is that $\neg[Y \leftarrow y]\varphi$ is equivalent to $[Y \leftarrow y]\neg\varphi$.

Ignore for now the requirements that the disjuncts be mutually exclusive, that all interventions be distinct, and that there be no leading formulas involving interventions. Using standard propositional reasoning, we can transform a formula $\varphi$ to an equivalent formula in DNF, where the literals are either simple formulas or intervention formulas (i.e., formulas of the form $[X \leftarrow x]\varphi$). Of course, the disjuncts may not be mutually exclusive. Again, using straightforward propositional reasoning, we can convert the formula to a DNF where the disjuncts are mutually exclusive. Rather than writing out the tedious details, we give an example. Consider a formula of the form $(\varphi_1 \wedge \varphi_2) \vee (\varphi_3 \wedge \varphi_4)$. This is propositionally equivalent to

$$(\varphi_1 \wedge \varphi_2 \wedge \varphi_3 \wedge \varphi_4) \vee (\varphi_1 \wedge \varphi_2 \wedge \neg\varphi_3 \wedge \varphi_4) \vee (\varphi_1 \wedge \varphi_2 \wedge \varphi_3 \wedge \neg\varphi_4) \vee (\varphi_1 \wedge \varphi_2 \wedge \neg\varphi_3 \wedge \neg\varphi_4)$$
$$\vee (\neg\varphi_1 \wedge \varphi_2 \wedge \varphi_3 \wedge \varphi_4) \vee (\varphi_1 \wedge \neg\varphi_2 \wedge \varphi_3 \wedge \varphi_4) \vee (\neg\varphi_1 \wedge \neg\varphi_2 \wedge \varphi_3 \wedge \varphi_4).$$

We can now apply the two equivalences mentioned above to remove leading negations from intervention formulas and to ensure that, in each disjunct, all interventions are distinct. These transformations maintain the fact that the disjuncts are mutually exclusive.

Since the disjuncts in $\psi$ are mutually exclusive, the probability of $\psi$ is the sum of the probabilities of the disjuncts; that is, $\Pr(\psi) = \sum_{i \in \{1,\ldots,m\}} \Pr(\psi_i)$. To compute the probability of a disjunct $\psi_i$, we first simplify it using the following two observations. First, if a formula involves an intervention $[X \leftarrow x]$ on some variable $X$ such that $X$ is also set to $x$ in the formula, such as $(X = x \wedge [X \leftarrow x, Z \leftarrow z](Y = 1))$, the intervention $X \leftarrow x$ is redundant and can be dropped; for example, $X = 0 \wedge [X \leftarrow 0, Z \leftarrow 1](Y = 1)$ is equivalent to $X = 0 \wedge [Z \leftarrow 1](Y = 1)$. Second, if an intervention formula does not contain a descendant of the intervened variables, such as $\psi = [\vec{X} \leftarrow \vec{x}](\psi_1 \wedge \psi_2)$, where all variables in $\psi_1$ are non-descendants of the variables in $\vec{X}$, then the variables in $\psi_1$ are not affected by the intervention, so $\psi_1$ can be pulled out of the scope of the intervention; that is, $\psi$ is equivalent to $\psi_1 \wedge [X \leftarrow x](\psi_2)$. Using these observations, we remove all interventions that are redundant and pull formulas involving only non-descendants of the intervened variables out of the intervention formula.

After this simplification, without loss of generality, the disjunct $\psi_i$ is a conjunction of formulas $\psi_{i0} \wedge \left( \bigwedge_{j \in \{1,\dots,r_i\}} \psi_{ij} \right)$, where $\psi_{i0} = \left( \bigwedge_{j \in \{1,\dots,s_i\}} (Z_{ij} = z_{ij}) \right)$ is a simple formula (with no intervention), and $\psi_{ij}$ for $j > 0$ has the form $[\vec{X}_j \leftarrow \vec{x}_j](\bigwedge_{k \in \{1,\dots,t_{ij}\}} Y_{ijk} = y_{ijk})$, where $Y_{ijk}$ is a descendant of some variable in $\vec{X}_j$ in $M$. The following theorem proves the result for $\psi_i$, which completes the proof.

**Theorem B.2.** *If $M$ is a CBN and $\psi_i = \psi_{i0} \wedge \left( \bigwedge_{j \in \{1,\dots,r_i\}} \psi_{ij} \right)$, where $\psi_i$ contains no redundant interventions, $\psi_{i0} = \left( \bigwedge_{j \in \{1,\dots,s_i\}} (Z_{ij} = z_{ij}) \right)$ is a simple formula (with no interventions), and $\psi_{ij}$ for $j > 0$ has the form $[\vec{X}_j \leftarrow \vec{x}_j](\bigwedge_{k \in \{1,\dots,t_{ij}\}} Y_{ijk} = y_{ijk})$, where $Y_{ijk}$ is a descendant of some variable in $\vec{X}_j$ in $M$, then $\Pr(\psi_i)$ can be computed by determining the probability of formulas that do not involve an intervention.*

*Proof.* The proof proceeds along lines very similar to the proof of Theorem 5.2.

Let $\bar{\mathcal{Z}} = \cup_{j=1}^{s_i} Z_{ij}$, $\mathcal{Z} = \mathcal{U} \cup \mathcal{V} \setminus \bar{\mathcal{Z}}$, and $\bar{z} = \{z_{ij} : j \in \{1,\dots,s_i\}\}$. $\mathcal{T}_{\mathcal{Z}}$ has $2^{|\mathcal{Z}|}$ settings. For a setting $c \in \mathcal{T}_{\mathcal{Z}}$, let $c_Z$ be the setting of the variable $Z$ in $c$. We use $Pa_A(Z)$ to denote $A \cap Pa(Z)$, i.e., the set of parents of $Z$ in $A$ and $\bar{z}_A$ to denote the values in $\bar{z}$ for all variables in $A$. Then

$$\Pr_M(\psi_i) =$$
$$\sum_{c \in \mathcal{T}_{\mathcal{Z}}} \Pr_M \left( \bigwedge_{j \in \{1,\dots,s_i\}} (Z_{ij} = z_{ij}) \wedge \left( \bigwedge_{j \in \{1,\dots,r_i\}} [\vec{X}_j \leftarrow \vec{x}_j]( \bigwedge_{k \in \{1,\dots,t_{ij}\}} Y_{ijk} = y_{ijk}) \right) \wedge \bigwedge_{Z \in \mathcal{Z}} (Z = c_Z) \right).$$

We next compute the probability of

$$\psi_{ic} = \left( \bigwedge_{j \in \{1,\dots,s_i\}} (Z_{ij} = z_{ij}) \wedge \left( \bigwedge_{j \in \{1,\dots,r_i\}} [\vec{X}_j \leftarrow \vec{x}_j]( \bigwedge_{k \in \{1,\dots,t_{ij}\}} Y_{ijk} = y_{ijk}) \right) \wedge \bigwedge_{Z \in \mathcal{Z}} (Z = c_Z) \right).$$

From Theorem 3.6, it follows that in all causal models $M'$ compatible with $M$,

$$\Pr_{M'}(\psi_{ic}) = \sum_{\varphi_\alpha \implies \psi_{ic}} \Pr_{M'}(\varphi_\alpha).$$

Now $\varphi_\alpha \implies \psi_{ic}$ holds whenever $\alpha$ contains the following events:

1. $(Z_{ij} = z_{ij} \mid Pa(Z_{ij}) = c_{Pa(Z_{ij})})$, for all $j \in \{1,\dots,s_i\}$;

2. $(Z = c_Z \mid Pa_{\mathcal{Z}}(Z) = c_{Pa_{\mathcal{Z}}(Z)}, Pa_{\bar{\mathcal{Z}}}(Z) = \bar{z}_{Pa_{\bar{\mathcal{Z}}}(Z)})$, for all $Z \in \mathcal{Z}$;

3. $(X = c_X^j \mid Pa(X) = c_{Pa(X)}^j)$, for all $X \in \mathcal{X}_j'$, where $\mathcal{X}_j'$ consists of all descendants of the intervened variables in $\vec{X}_j$ other than the variables in $\vec{X}_j$ and $c^j \in \mathcal{T}_j'$, the set of settings of the variables in $\mathcal{U} \cup \mathcal{V}$, where the following variables are fixed as follows:

   (a) $\vec{X}_j = \vec{x}_j$,
   (b) $Y_{ijk} = y_{ijk}$ for all $k \in \{1,\dots,t_{ij}\}$,
   (c) $Z_{ik} = z_{ik}$ for $Z_{ik} \notin (\vec{X}_j \cup \mathcal{X}_j')$, $k \in \{1,\dots,s_i\}$,
   (d) $Z = c_Z$ for all $Z \in \mathcal{Z}$ and $Z \notin (\vec{X}_j \cup \mathcal{X}_j')$.

Intuitively, $\mathcal{T}_j'$ captures all possible post-intervention settings of all variables that are descendants of $\vec{X}_j$, while fixing $Y_{ijk}$s as $y_{ijk}$. By fixing the third set of events, $(X = c_X^j | Pa(X) = c_{Pa(X)}^j)$ for all $X \in \mathcal{X}_j'$, we ensure that all events involving descendants of $\vec{X}_j$ are consistent with respect to one of the post-intervention settings $c^j \in \mathcal{T}_j'$. These events represent the effects of interventions in $\vec{X}_j \leftarrow \vec{x}_j$ on its descendants. For example, consider a causal graph as shown below and $\psi_i = [X_1 \leftarrow 1, X_3 \leftarrow 1](Y = 1)$.

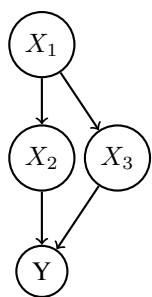

In this case, $\vec{X}_j = \{X_1 \leftarrow 1, X_3 \leftarrow 1\}$. By the conditions mentioned above, $\alpha$ must contain one of the two events $((Y = 1 \mid X_2 = 0, X_3 = 1) \wedge (X_2 = 0 \mid X_1 = 1))$ or $((Y = 1 \mid X_2 = 1, X_3 = 1) \wedge (X_2 = 1 \mid X_1 = 1))$, because $\mathcal{T}'_1 = \{\{X_1 = 1, X_2 = 0, X_3 = 1, Y = 1\}, \{X_1 = 1, X_2 = 1, X_3 = 1, Y = 1\}\}$. This condition ensures that if $X_1 = 1$ and $X_3 = 1$, then $\varphi_\alpha$ implies $Y = 1$. It is easy to see that if $\alpha$ does not contain either of these two events, then it must contain $((Y = 0 \mid X_2 = 0, X_3 = 1) \wedge (X_2 = 0 \mid X_1 = 1))$ or $((Y = 0 \mid X_2 = 1, X_3 = 1) \wedge (X_2 = 1 \mid X_1 = 1))$, in which case $\varphi_\alpha$ does not imply $\psi_i$.

Let $S_c$ consist of all ccces that contain these events, and let $\varphi_{S_c}$ be the conjunction of the formulas corresponding to the events in $S_c$. Thus,

$$
\begin{aligned}
\varphi_{S_c} = & \left( \bigwedge_{j' \in \{1, \ldots, s_i\}} [Pa(Z_{ij'}) \leftarrow c_{Pa(Z_{ij'})}](Z_{ij'} = z_{ij'}) \right) \\
& \wedge \left( \bigwedge_{Z \in \mathcal{Z}} [Pa_{\mathcal{Z}}(Z) \leftarrow c_{Pa_{\mathcal{Z}}(Z)} \wedge Pa_{\bar{\mathcal{Z}}}(Z) \leftarrow \bar{z}_{Pa_{\bar{\mathcal{Z}}}(Z)}](Z = c_Z) \right) \\
& \wedge \bigwedge_{j \in \{1, \ldots, r_i\}} \left( \bigvee_{c^j \in \mathcal{T}'_j} \left( \bigwedge_{X \in \mathcal{X}'_j} [Pa(X) \leftarrow c^j_{Pa(X)}](X = c^j_X) \right) \right) \\
= & \left( \bigwedge_{j' \in \{1, \ldots, s_i\}} [Pa(Z_{ij'}) \leftarrow c_{Pa(Z_{ij'})}](Z_{ij'} = z_{ij'}) \right) \\
& \wedge \left( \bigwedge_{Z \in \mathcal{Z}} [Pa_{\mathcal{Z}}(Z) \leftarrow c_{Pa_{\mathcal{Z}}(Z)} \wedge Pa_{\bar{\mathcal{Z}}}(Z) \leftarrow \bar{z}_{Pa_{\bar{\mathcal{Z}}}(Z)}](Z = c_Z) \right) \\
& \wedge \left( \bigvee_{\{c^j \in \mathcal{T}'_j \, : \, j \in \{1, \ldots, r_i\}\}} \left( \bigwedge_{X \in \mathcal{X}'_l, \, l \in \{1, \ldots, r_i\}} [Pa(X) \leftarrow c^l_{Pa(X)}](X = c^l_X) \right) \right).
\end{aligned}
$$

Then by Theorem 3.6,

$$
\varphi_{S_c} \Leftrightarrow \bigvee_{\alpha \in S_c} \varphi_\alpha.
$$

Since the formulas $\varphi_\alpha$ for distinct ccces in $S_c$ are mutually exclusive, we have that $\Pr_{M'}(\psi_{ic}) = \Pr_{M'}(\varphi_S)$. Therefore,

$$
\begin{aligned}
& \Pr_{M'}(\psi_i) \\
= & \sum_{c \in \mathcal{T}_{\mathcal{Z}}} \Pr_{M'}(\psi_{ic}) \\
= & \sum_{\substack{c \in \mathcal{T}_{\mathcal{Z}}, \\ c^j \in \mathcal{T}'_j \, : \, j \in \{1, \ldots, r_i\}}} \Pr_{M'} \left( \bigwedge_{j' \in \{1, \ldots, s_i\}} (Z_{ij'} = z_{ij'}) \wedge \bigwedge_{Z \in \mathcal{Z}} (Z = c_Z) \right. \\
& \hspace{4cm} \left. \wedge \bigwedge_{X \in \mathcal{X}'_l, \, l \in \{1, \ldots, r_i\}} [Pa(X) \leftarrow c^l_{Pa(X)}](X = c^l_X) \right).
\end{aligned}
$$

We can further simplify this expression. Specifically, we can get rid of $[Pa(X) \leftarrow c^j_{Pa(X)}](X = c^j_X)$ for all descendants $X$ of some $Y_{ijk}$ with $k \in \{1, \ldots, t_{ij}\}$ and $j \in \{1, \ldots, s_i\}$. We leave the details to the reader. The expression above may be infeasible for some combinations of settings $c \in \mathcal{T}_{\mathcal{Z}}$ and $c^l$ for all $l \in \{1, \ldots, r_i\}$. For example $[X \leftarrow 1](Y = 0) \wedge [X \leftarrow 1](Y = 1)$ has zero probability. Furthermore, certain formulas in $\bigwedge_{X \in \mathcal{X}'_l, l \in \{1, \ldots, r_i\}} [Pa(X) \leftarrow c^l_{Pa(X)}](X = c^l_X)$ may be duplicates, and some interventions may be redundant. We need to drop the duplicates and

redundant interventions before further simplifying the expression. For ease of exposition, we assume that the expression is feasible, all conjuncts in $\bigwedge_{X \in \mathcal{X}'_l, l \in \{1, \ldots, r_i\}} [Pa(X) \leftarrow c^l_{Pa(X)}](X = c^l_X)$ are distinct, and all interventions are non-redundant.

If $M'$ is i-compatible with $M$, then we can further conclude that

$$
\begin{aligned}
&\mathrm{Pr}_{M'}(\psi_i) \\
&= \sum_{\substack{c \in \mathcal{T}_{\mathcal{Z}}, \\ c^j \in \mathcal{T}'_j : j \in \{1, \ldots, r_i\}}} \left( \mathrm{Pr}_{M'} \left( \bigwedge_{j' \in \{1, \ldots, s_i\}} (Z_{ij'} = z_{ij'}) \wedge \bigwedge_{Z \in \mathcal{Z}} (Z = c_Z) \right) \right. \\
&\qquad\qquad\qquad \left. \times \mathrm{Pr}_{M'} \left( \bigwedge_{\substack{X \in \mathcal{X}'_l, \\ l \in \{1, \ldots, r_i\}}} [Pa(X) \leftarrow c^l_{Pa(X)}](X = c^l_X) \right) \right) \\
&= \sum_{\substack{c \in \mathcal{T}_{\mathcal{Z}}, \\ c^j \in \mathcal{T}'_j : j \in \{1, \ldots, r_i\}}} \left( \mathrm{Pr}_{M'} \left( \bigwedge_{j' \in \{1, \ldots, s_i\}} (Z_{ij'} = z_{ij'}) \wedge \bigwedge_{Z \in \mathcal{Z}} (Z = c_Z) \right) \right. \\
&\qquad\qquad\qquad \left. \prod_{\substack{X \in \mathcal{X}'_l, \\ l \in \{1, \ldots, r_i\}}} \mathrm{Pr}_{M'}(X = c^l_X \mid Pa(X) \leftarrow c^l_{Pa(X)}) \right).
\end{aligned}
$$

Since $\mathrm{Pr}_{M'}(\psi_i) = \mathrm{Pr}_M(\psi_i)$, we get the desired result. $\qquad\square$

In terms of complexity, each intervention $\vec{X}_j \leftarrow \vec{x}_j$ requires at most $2^n$ different settings in the set $\mathcal{T}'_j$. Therefore, the expression above for $\mathrm{Pr}_M(\psi_i)$ has $O(2^{n(r_i+1)})$ setting combinations in the summation and $O(nr_i+1)$ conditional probability calculations for each such setting. This shows that an arbitrary formula $\psi$ can be evaluated in terms of $O(m(nr^* + 1)2^{n(r^*+1)})$ conditional probability calculations, where $r^*$ is the maximum number of conjuncts in a disjunction $\psi_i$ that involve at least one intervention, and $m$ is the number of disjuncts in the DNF.

