# OpenReview forum: "Intervention and Conditioning in Causal Bayesian Networks"
_NeurIPS.cc/2024/Conference — NeurIPS 2024 poster_

### Official Review · Reviewer_3ny6 · 2024-07-08

**Soundness:** 2
**Presentation:** 2
**Contribution:** 2
**Rating:** 5
**Confidence:** 3

**Summary:**

This paper builds off recent work showing that Pearl's proposed approach for calculating conditional probabilities involving interventions is incorrect.  Using Pearl's concept of autonomy (independence of mechanisms), the authors build a series of formalisms to calculate arbitrary conditional probabilities in causal Bayesian networks and show how these formalisms relate CBNs and functional causal models.

**Strengths:**

I think this paper is overall well-organized and written.  While there is a lot of notation and terminology, the definitions are presented clearly, and the narrative is relatively easy to follow.

**Weaknesses:**

There are many cases of imprecise language or strange descriptions that, at least on the surface, appear incorrect.  These vary in severity, but their frequency is worrying.
- line 17: "in epidemiology, causal models are instrumental in deciphering the relationships between lifestyle choices and health outcomes (Greenland, Pearl, and Robins 1999)".  I'm not sure that this is true, or, if it is (my knowledge of current epidemiological practice is limited), that citation certainly doesn't support this claim.  The abstract in the cited paper says that "Causal diagrams can provide a starting point", which is a far cry from them being "instrumental".
- line 33: "In a causal model [...] there is a deterministic (structural) equation, that gives the value of Y as a function of the values of its parents." There are a few issues here.  First, the previous line does introduce the functional causal model vs CBN distinction, but you haven't yet said that you'll be using the phrase "causal models" exclusively to refer to "functional causal models", so readers will assume you mean both types.  However, CBNs are not deterministic, so this line just reads as incorrect.  Second, even if we assume that you're using 'causal model' here as you later define it in line 123, it's still incorrect since line 123 says that "in the rest of this paper, when refer to a 'causal model', we mean a probabilistic functional causal model".  I discuss this issue in more detail below, but the terminology around 'functional causal models' needs to be reworked to be less confusing.
- line 45-46: "Pearl assumes that the mechanisms that determine how interventions work (which are given by the cpts in the case of CBNs and the structural equations in the case of causal models), [...]" I understand that interventions are performed by replacing the CPT with a constant for CBNs or by replacing the right hand side of the equation in SEMs.  However, I wouldn't say that the CPT or the structural equations are "the mechanism that determine how interventions work".  Sort of the opposite - they're the mechanisms that determines how the model operates under observational conditions.  The intervention makes change to a CPT or structural equation, but nothing in that table or equation is determining how the intervention works...
- line 108: "if a theory is recursive, [...]" This is the only time you refer to the noun 'theory', and in line 105, you define the term recursive wrt a model, not a theory.  Do you just mean model here?
- line 127: "we focus on recursive CBNs that can be characterized by a dag".  This is strange and redundant.  Assuming here, recursive == acyclic, as you defined earlier, all CBNs are acyclic, so the phrase "recursive CBN" is redundant.  Similarly with "that can be characterized by a dag" - Bayes nets use a DAG representation, so all CBNs can be characterized by a DAG. (am I missing some nuance you're trying to convey here?)
- line 155: Until this point, you've been using $\phi$ to refer a specific formula (e.g., at the beginning of line 155).  Then, lines 155-156 use $\psi$ before returning to $\phi$ again.  Is this a typo?  If not and there's a meaningful difference, you need to describe that.
- line 161: I think this is just a typo.  The line as-is reads $(M_{\vec{Y} \leftarrow \vec{y}}, \vec{u})_{\vec{Y} \leftarrow \vec{y}}$.  However, the text afterwards refers to

$\vec{u}_{\vec{Y} \leftarrow \vec{y}}$.  So should the initial line instead be

$(M_{\vec{Y} \leftarrow \vec{y}}, \vec{u}_{\vec{Y} \leftarrow \vec{y}})$ ? (excuse the weird line breaks...it didn't like multiple equations in one line for some reason)
- footnote 5: "it is conceptually somewhat inconsistent to allow interventions on exogenous variables, since their value is assumed to be determined by factors outside the model, [...]" I agree it's conceptually strange to reason about intervening on exogenous variables, but not for that reason.  After all, in practice, we rarely observe all causes of endogenous variables (e.g., in a healthcare setting, many subtle biological factors will influence a patient's status, while only some biological indicators can be/are actually measured), so their values are, at least partially determined by factors outside the model.  The conceptual mismatch seems more rooted in the inability to influence those features or their status as logically outside the system under study.
- line 180: "The situation gets worse [...]", where the preceding paragraph discusses causal formulas with interventions.  However, that paragraph didn't really discuss anything as being bad or problematic, so it's unclear what situation is getting worse, or what getting "worse" even means in this context.
- lines 205-207: "Explicitly mentioning the CBN M is necessary, since on the right-hand side of the conditional with left-hand side Y, we have all the parents of Y; what the parents are depends on M." However, the conditional being referred to here doesn't use notation like Pa(Y) (if it did, I would understand and agree with this point).  The conditional instead lists out "$X_1 = x_1 ... X_m = x_m$, with those, in this case, corresponding to the parents of Y.  However, I don't see anything in the form of the conditional that requires that the right-hand side exclusively on the parents of the left-hand side.  If it's listing the variables out explicitly, there's no problem specifying "CBN M", but I don't see why it's necessary.
- line 257-258: "$Z = 1 | X = 0, Y = 0$ as saying 'if X were (set to) 0 and Y were (set to) 0, then Z would be 1" My issue here is with the phrase "set to".  "Set to" implies intervention, but that would use an arrow, not an =.  I assume what this notation means if "if X were (observed at) value 0 and Y were (observed at) value 0, then Z would be 1"


The authors make a distinction throughout this paper between "causal Bayesian networks" and "functional causal models".  This is a useful distinction, but it's muddied by the authors' strange decision to simply call functional causal models "causal models", creating a frequent contrast between CBNs and causal models that implies that CBNs are not causal models.  This is even stranger in light of line 140, where the authors say "a causal model can be viewed as a CBN", suggesting that CBNs are a broader class....and yet they're somehow not 'causal models'??  I think this is just a terminology issue, but it's frequently confusing - I'd either find another term, say "functional causal model" throughout, or abbreviate it to FCM.
- at the very least, make the notation consistent.  You don't actually say that "causal model" == "functional causal model" explicitly until line 123.  However, before that point, you switch between saying "functional causal model" (lines 31, 37, 41, 44), "causal model" (32, 47, 53, 72, 74...), and "(functional) causal model (78).  I'm also unclear if the use of "causal model" in the introductions first two paragraphs are intended to only refer to "functional causal models".  I'm guessing not, but it's unclear when the transition takes place.

Please correct me if I'm wrong, but the motivation for this paper seems to be based on the findings of Beckers (2023) (since they show that Pearl's calculations are incorrect, this prompting the need for this paper to show how to do those calculations correctly).  However, looking for Beckers 2023, it's cited as an unpublished manuscript, so I'm not sure how to evaluate the motivation for this paper.  The authors do provide an example, presumably based on Beckers 2023, in footnote 1.  However, the example is strange (3 binary variables, two of which have deterministic dependencies) and the example query P(X = 1 | X = 1) seems nonsensical.  The authors don't go into any more detail about why Pearl's method for calculating probabilities is insufficient, they only give a strange toy example with deterministic dependencies, and I'm unable to look at the paper this work is based on (at least, I was unable to find it when looking at Beckers website); given this, it makes it very hard for me to assess the contribution of this paper since I can't understand the motivation and what problem it's trying to solve.

The biggest piece I'm missing, and the primary contributor towards my low score (along with the fuzziness around the Beckers result), is the lack of motivation for this work.  The authors say that there's a weakness in CBN theory (as exposed by Beckers 2023) and present a series of formalisms to calculate probabilities that over comes this weakness.  However, the negative effects of this weakness are never shown (apart from a toy example in a footnote), leaving the whole paper seeming like an interesting, but ultimately unimportant, mathematical exercise.  I don't mean that to come across as overly harsh, and I don't necessarily believe that this contribution is unimportant.  However, the authors have failed to show any evidence to the contrary.  If the authors can (1) discuss the limitations and pitfalls of using the calculations from Pearl and (2) provide a grounded example where such limitations could pose problems for causal estimation, I'm more than happy to raise my score.

**Questions:**

Is the unpublished Beckers 2023 manuscript available anywhere?  If not, can you describe the main argument of that work?

What is the practical use case of this work?  The only examples provided are toy examples, which, while useful for explaining the approach, don't do much for showing how the method presented in the paper could be used in a real-world setting to answer a relevant causal query.

In line 392, is the $f$ on the left-hand side of the equation the same as the $f$ on the right-hand side?  You define $f$ here as "the value of $U_Y$", and it seems strange to me that $U_Y$ would be an equation that depends on $x_1, ..., x_n$.  However, I'm not otherwise sure why you'd introduce a new variable that shares the name with a function in the same equation.

**Limitations:**

I'm unclear about the applicability and use case of this work, so I'm not sure what the limitations could be.


I've raised my score from a 3 to a 5.  If the other reviewers don't view the fact that this work is motivated by fairly significant findings that cannot be viewed and assessed at the time of publication, then I'm completely fine with accepting.  The main sticking point for me is not being able to assess the severity of the problem that this is solving.

---

> ### Author Rebuttal · Authors · 2024-08-01
>
> lines 17 and 33: we will clarify the issues you mentioned.
>
> lines 45-46: We think there's a natural reading of this material this
> is consistent with what we said.  Suppose that you intervene to set
> X=1. To find out the effect of this, you consult the structural
> equation.  That makes the equation the mechanism that determines how
> the intervention works.
>
> l. 108: We did mean recursive model.
>
> l. 127: Although CBNs are typically taken to be acyclic, there is
> certainly work on dealing with cyclic CBNs and BNs.  (See, for example,
> Baier et al., ON the foundation of cycles in Bayesian network,
> Lecture Notes in Computer Science vol. 13360, pp. 343-363.)
>
> l. 155: good catch!  This is a typo
>
> l. 161: indeed, this is also a typo
>
> footnote 5: We seem to have quite different views of exogenous
> variables. We are willing to assume
> that (at least, in some applications) there are reasonable models
> where you can observe all causes of endogenous variables.  It seems
> (to us) strange to limit the applicability of causal models to those
> applications where this is not the case.  That said, we agree that an
> agent in the system typically can't influence the value of an
> exogenous variable.
>
> l. 180: We'll clarify.  We meant that it gets harder to assign a
> probability to a formula.
>
> l. 205: Even if we don't use the notation Pa(Y), we still have to
> explicitly list all the parents of Y in M. Different choices of M
> will in general involve different parents and hence result in different
> formulas.  So we do have to list the M explicitly.
>
> l. 257: "Set to" is closer to what we meant here.  We just made up
> this notation temporarily to define what we really care about.  That
> said, you're right that we probably should have used <- rather than =.
>
> - Using FCM is a good suggestion.
>
> Question 1: Our paper is not really based on the findings of Beckers.
> The only
> fact that we use from Beckers' paper is his observation that Pearl's
> procedure gives incorrect results.  (Beckers does an analysis of why
> the procedure is incorrect, and gives a more general example.)
> We strongly disagree that the
> example is nonsensical.
> Pearl's procedure can clearly be applied to the
> formula X=1 in the model constructed, and gives the answer P(X=1|X=1)
> = 1/2, which is clearly incorrect.  It is up to the user of the
> procedure to explain when it can (and can't) be applied.
> Mathematically, one counterexample is sufficient
> to disprove a result. Beckers
> shows that the problem is rather deep, but it seemed inappropriate
> to steal his thunder, so to speak, by including this material in our
> paper.  Unfortunately, the paper is still an early draft, and is not
> available on the web.
>
>
> Roughly speaking, Beckers shows that the problem is that the first step
> in Pearl's procedure
> applies conditioning only to the exogenous variables.  It should be
> applied to the joint
> distribution on exogenous and endogenous variables.  While the two
> approaches agree in
> functional causal models, they do not agree in CBNs.  On the other hand,
> applying conditioning
> to the joint distribution typically breaks the independencies that are
> required for having a Markov
> factorization.
>
>
> Question 2: We will give examples of the usefulness of our
> approach in the full paper.  We agree that this is important.
>
> Question 3: The f is the same on both sides of the equation. U_Y is a
> random variable (not an equation), and its values are functions. F_Y
> gets as input a value of each of the variables X_1, ..., X_n, and U_Y.
> Given those values, it computes a value of Y. Since f is a function
> mapping values of X_1, ..., X_n to a value of Y, this equation does
> the right thing.

---

> > ### Comment · Reviewer_3ny6 · 2024-08-08
> >
> > I appreciate the authors' response, which go part of the way to answering my concerns.
> >
> > To your point that your paper isn't really based on the findings of Beckers, maybe 'based' isn't the right word, but at the very least it seems somewhat dependent on the findings of Beckers.  The message I get from the introduction as it stands is "We want to calculate probabilities under intervention and conditioning.  The classic method of doing that in CBNs doesn't actually work, as shown by Beckers.  Here, we show a way to do that which actually works."  If that's not what I should be getting from the introduction, then it should be reworked to show the value of this work independent of whether or not Pearl's CBN procedure is correct.
> >
> > Given that current understanding, I agree with Reviewer nbxs (Reviewer 2?) that a non-toy example for Footnote 1 would help a lot.  I understand and appreciate that the authors don't want to steal Beckers' thunder.  However, as it stands, the paper does nothing to suggest to me the seriousness of what Beckers has found.  The fact that Footnote 1 only has 3 variables isn't really a problem.  The problem for me is that of those 3 variables, 2 of them (U and Y) are exactly the same ($Y = U$), but somehow one is exogenous and one is endogenous.  And then the query we're asking is $p(X = 1 | X = 1)$, a query that I struggle to see the practical use of.  It's possible that the incorrect result seen in Footnote 1 would be present in a wide range of situations, but with this as the provided example and no access to Beckers' paper, my first thought is "So does it only break in extreme fringe cases like this?"  I understand that one counter-example is enough to show that it's incorrect.  However, there's a big practical difference between "It will often given incorrect probabilities in a wide range of practical situations" and "It will break if we have deterministic dependence spanning exogenous-endogenous edges and ask then ask a query where the left and right hand sides are exactly the same".
> >
> > > "Question 2: We will give examples of the usefulness of our approach in the full paper. We agree that this is important."
> >
> > Would it be possible to provide an example now (or at least a sketch of what this will look like)?

---

> > > ### Author Response · Authors · 2024-08-09
> > > **Response to the first question about the counterexample**
> > >
> > > Thanks again for your comments.
> > >
> > > > To your point that your paper isn't really based on the findings of
> > > > Beckers, maybe 'based' isn't the right word, but at the very least it
> > > > seems somewhat dependent on the findings of Beckers.
> > >
> > > The paper was certainly motivated by Beckers' observation, but is
> > > not dependent on it.  That said, once we appreciated Beckers'
> > > observation, our reaction was pretty much what you said: "We want to
> > > calculate
> > > probabilities under intervention and conditioning. The classic method
> > > of doing that in CBNs doesn't actually work, as shown by
> > > Beckers. Here, we show a way to do that which actually works."
> > >
> > > So, yes, that's what you should be getting from the introduction!
> > >
> > > > Given that current understanding, I agree with Reviewer nbxs (Reviewer
> > > > 2?) that a non-toy example for Footnote 1 would help a lot. I
> > > > understand and appreciate that the authors don't want to steal
> > > > Beckers' thunder. However, as it stands, the paper does nothing to
> > > > suggest to me the seriousness of what Beckers has found. The fact that
> > > > Footnote 1 only has 3 variables isn't really a problem. The problem
> > > > for me is that of those 3 variables, 2 of them (U and Y) are exactly
> > > > the same ($Y = U$), but somehow one is exogenous and one is
> > > > endogenous. And then the query we're asking is $p(X = 1 | X = 1)$, a
> > > > query that I struggle to see the practical use of. It's possible that
> > > > the incorrect result seen in Footnote 1 would be present in a wide
> > > > range of situations, but with this as the provided example and no
> > > > access to Beckers' paper, my first thought is "So does it only break
> > > > in extreme fringe cases like this?" I understand that one
> > > > counter-example is enough to show that it's incorrect. However,
> > > > there's a big practical difference between "It will often given
> > > > incorrect probabilities in a wide range of practical situations" and
> > > > "It will break if we have deterministic dependence spanning
> > > > exogenous-endogenous edges and ask then ask a query where the left and
> > > > right hand sides are exactly the same".
> > >
> > > We do agree with the point that it's useful for the reader  to
> > > appreciate how deep the problem is.  But, at the same time, as we said,
> > > we feel it inappropriate to steal Beckers' thunder.  We're also not
> > > sure that any single example will solve the problem that you raise.
> > > But we could add the following modification of the example of the
> > > example in the footnote: Suppose that Y = U with probability 1/2 and Y
> > > = 1-U with probability 1/2. Similarly X = Y with probability 1/2 and 1-Y
> > > with
> > > probability 1/2. Even in this case, Pr(X=1|X=1) would evaluate to
> > > 1/2.  This shows that dependence between exogenous and endogenous
> > > variables is not the problem. Rather, whenever the exogenous
> > > distribution does not change by conditioning, it is easy to construct
> > > a counterexample.  Would adding that help?  Perhaps what might work
> > > best is (with Beckers' agreement) to add a few sentences of
> > > discussion which we clearly indicate are taken from Beckers' paper
> > > that explains why he thinks that this is a deep problem, that isn't
> > > easily fixable.

---

> > > > ### Author Response · Authors · 2024-08-09
> > > > **Example applications**
> > > >
> > > > > "Question 2: We will give examples of the usefulness of our
> > > > > approach in the full paper. We agree that this is important."
> > > >
> > > > > Would it be possible to provide an example now (or at least a sketch of
> > > > > what this will look like)?
> > > >
> > > > Sure!  Generally, any setting where exogenous variables can be observed
> > > > would directly benefit from our results. We present two diverse example
> > > > scenarios here.
> > > > (i) While debugging systems (ml pipelines, database engines
> > > > or any general software) and network failures,
> > > > users have access to all parameters
> > > > related to the code and the execution environment [1,2].
> > > > With this information, a causal graph over different
> > > > blocks of code, its parameters and other environment
> > > > variables like information about background processes
> > > > can be constructed.
> > > > This means we can address queries like "Given that
> > > > component A is faulty, with what probability would repairing component
> > > > B solve the problem", using our techniques.
> > > >
> > > >
> > > > (ii) Manufacturing pipelines across various industries,
> > > > such as semiconductor fabrication, pharmaceuticals,
> > > > automobile assembly, and battery production, typically
> > > > consist of a series of interconnected stages. Each of these
> > > > stages is equipped with several sensors designed to monitor
> > > > and measure critical environmental conditions that directly
> > > > impact the production process. These sensors collect data on
> > > > variables such as temperature, humidity, pressure, and other
> > > > factors that can influence the quality, efficiency, and consistency
> > > > of the final product. For instance, in semiconductor fabrication,
> > > > precise control of environmental conditions like temperature and
> > > > humidity is crucial to ensuring the integrity of the microchips
> > > > produced [3]. Similarly, in pharmaceutical manufacturing, sensors monitor
> > > > parameters like pH levels and chemical concentrations to maintain
> > > > the efficacy of the drugs being produced.
> > > > Thus, we can address queries with what probability would a
> > > > temperature increase of 3 degrees Celsius result in a poor quality product,
> > > > given that the humidity is high.
> > > >
> > > >
> > > > Other potential applications of our framework include
> > > > (a) modelling player performance in sports by considering factors like
> > > > injury, skill, sports facilities, etc.
> > > > (b) urban planning scenarios to analyze the impact of zoning laws, interest
> > > > rate and other factors on house prices
> > > > (c) modelling agriculture yield by considering variables like
> > > > soil quality, weather conditions.
> > > > (d) transportation applications like railway operation management,
> > > > where data is collected from different sensors and monitoring devices
> > > > about track integrity, train speed, braking performance,
> > > > and environmental variables such as temperature and precipitation.
> > > > In this case, environment information is exogenous as it can not be
> > > > intervened on.
> > > >
> > > > As can be seen, there is no lack of applications!

---

> > > > > ### Comment · Reviewer_3ny6 · 2024-08-13
> > > > >
> > > > > I appreciate the authors' clarifying comments and examples and apologize for not responding sooner.  This paper is in a weird spot for me since a lot of the motivation is currently unavailable, and it's unclear when Beckers' paper will be made public.  If the timing is just off by a couple of months, and Beckers' findings should be available publicly by NeurIPS or shortly thereafter, then this is less of an issue.  However, if Beckers' findings will still be an unpublished and unavailable manuscript a year from now, then this publication seems premature.
> > > > >
> > > > > Those concerns aside, I think the authors did address some of my concerns adequately and, with their proposed modifications, I'm happy to raise my score.

---

> > > > > > ### Author Response · Authors · 2024-08-13
> > > > > >
> > > > > > We will check with Beckers about the timeline. We'll also add a
> > > > > > discussion to the paper (from Beckers' paper) explaining why the problem
> > > > > > is deeper than just a counterexample. We thank you for
> > > > > > engaging with us and appreciate your willingness to raise your
> > > > > > score!

---

### Official Review · Reviewer_6Tw9 · 2024-07-12

**Soundness:** 4
**Presentation:** 3
**Contribution:** 4
**Rating:** 8
**Confidence:** 3

**Summary:**

The paper introduces a condition under which the counterfactual probabilities can be computed from a Causal Bayesian network (CBN). This is not the case in general and functional causal models such as SCMs are often required for counterfactual reasoning. Specifically, the paper shows that when the outcomes are independent under different parent instantiations (in addition to the independence of CPTs), counterfactual queries can also be answered with a CBN.

**Strengths:**

I found the main idea of the paper very intuitive and significant. The paper also provided many examples to make it easy to follow. I found the discussion on the conversion of CBNs to SCMs in Section 4 quite helpful, and it is interesting to see how this is translated into the independence assumptions within CPTs (main contribution in Section 3.2). The results in this paper allow us to answer counterfactual queries (counterfactual layer in PCH) with the information from the interventional layer, which can be quite useful in practice as well.

**Weaknesses:**

I think the paper can be more convincing if the authors could provide some real-world scenarios in which the CPTs satisfy the independence assumptions and in which they do not.

**Questions:**

The results make sense to me and I don't have any specific questions for now.

**Limitations:**

OK.

---

> ### Author Rebuttal · Authors · 2024-08-01
>
> Thank you for your positive comments.  We
> will provide more examples where the independence assumptions do and
> do not hold.

---

### Official Review · Reviewer_nbxs · 2024-07-12

**Soundness:** 3
**Presentation:** 1
**Contribution:** 2
**Rating:** 4
**Confidence:** 3

**Summary:**

This paper shows the identifiability of interventional formulas in Causal Bayesian Networks (CBNs) under an independence-of-machanism assumption. It also formalizes the construction of Balke and Pearl (1994) of how to convert a CBN to a Structural Causal Model (SCM).

**Strengths:**

1. This paper considers an important problem of identifying counterfactual formulas in CBN.
2. Statements made in the paper are rigorously proved.

**Weaknesses:**

My first concern is regarding the assumptions required to identify the probability queries in a CBN. As far as I comprehend, there are three assumptions involved

- The outcomes of opts for different variables are independent.
- For the opt of a single variable $Y$, the outcomes under different settings of $Y$'s parents are independent.
- We have available all the basic ccce formulas.

The first asm. is okay since it is just a restatement of Pearl's autonomous mechanism. The third asm. is also acceptable (thought can be costly for computation) since it just states that we need a full observation of the system's dynamics.

What really concerns me is the second asm., which basically requires that the system does not contain unobserved, exogenous variables. For example, if we have $Y\leftarrow U, Y\leftarrow X$, where $U$ is a unobserved, exogenous variable, it is hard to imagine why $Y|X=0$ and $Y|X=1$ can be independent (they are both correlated to $U$). The authors should provide more justification of this assumption, and some concrete, real-world examples where this asm. can hold.

My second convern is clarity. This paper involves many definitions and notations, so the author should pay special attention to clarity. For example, what is the problem of existing framworks? what are your assumptions and their rationalities? what are your core contributions? These contents should be stated clearly and explained in depth. It would be nice to put the discussions right after the assumptions/theorems rather than in a separate section.

Moreover, I feel a disconnection between the discussion of CBN and that of SCM. Now that your main result is the identification quantities in CBN, why do you mention SCM? Indeed, Sect. 4 connects CBN and SCM, but it is basically a refinement of Pearl's method and the novelty is limited. I would suggest you reconsider this part and save more space to discuss your main results.

**Questions:**

1. Can you give a nontrial, meaningful example where Peal's "abduction, action, and prediction" procedure fails (footnote 1)?
2. In line 176, why assigning causal formulas in CBN is hard? Considering the fact that we have defined intervention in CBN (line135), why we can not just apply the same definition as in line 160? I do not really understand your explanation in lines 354-361.

**Limitations:**

I would suggest the authors to include their limitations (e.g., assumption, computational cost) in the paper.

---

> ### Author Rebuttal · Authors · 2024-08-01
>
> - With regard to the second assumption needed to identify the
> probabilities in a CBN, we agree that it is nontrivial.  We will
> expand the discussion on lines 55-63 and provide more examples.
> Given our view of exogenous variables are simply variables whose
> values are determined "from the outside", such examples are easy to
> generate.  To take just one of many examples, climate conditions
> (temperature, sunlight, humidity, CO2 level)
> are often measured to help model crop yield in agriculture scenarios.
> These variables typically cannot be intervened on, and their values come
> from "outside the model", so they are best viewed as exogenous.  But they
> satisfy our independence assumptions.
>
> - We didn't really understand your second concern (although are happy
> to do what we can to address it once we do understand it).
> Specifically, we don't think that there's a problem with the
> existing frameworks.  We show that by making certain
> independence assumptions (which we thought were stated clearly), we
> are able to identify probabilities in a CBN.  We would be happy to
> give more examples to motivate our assumptions and show their
> applicability.
>
>
> We believe that the discussion of SCMs is important.  Like Pearl, we
> need to go through SCMs to get our result.  Indeed (given our
> independence assumptions), we *define* the probability of a formula in
> a CBN to be its probability in any one of a set  of SCMs.  So we can't
> even state our result without going through SCMs.  And while there is
> certainly some overlap between our approach and Pearl's, there are
> also some nontrivial differences. However, we will expand
> on the points mentioned by the reviewer to clarify the key insights and
> assumptions.
>
> Question 1: The example in footnote 1 is a simple scenario to demonstrate
> that Pearl's procedure violates basic conditional probability
> calculations.
> One counterexample is sufficient to prove that the three step
> procedure fails.
> Also, the example can be complicated by adding additional
> interventional terms.
> It is a toy example because (i) it is not the contribution of this paper.
> (ii) we wanted to convey the intuition with the simplest setting.
>
> Question 2:  In probability
> theory, we can only assign probability
> to events (subsets of the
> sample space).  So to assign a probability to a formula like [Y ←
> y]\phi in a CBN, we have to identify a sample space and an event
> corresponding to this formula in the sample space. We will add more
> justification to the paper.

---

> > ### Comment · Reviewer_nbxs · 2024-08-08
> >
> > I thank the authors for responding, I will maintain my score.

---

### Official Review · Reviewer_Yecn · 2024-07-12

**Soundness:** 3
**Presentation:** 2
**Contribution:** 3
**Rating:** 6
**Confidence:** 3

**Summary:**

They define counterfactual probability for causal Bayesian networks as the probability of i-compatible causal models.

In the appendix, they show how to calculate counterfactual probabilities and necessary/sufficient probabilities.

**Strengths:**

One needs definition like in this paper before doing any kind of research into counterfactuals

**Weaknesses:**

Exponential complexity

Lots of things in the appendix

Some proofs are hard to understand

A lot of the motivation depends on an unpublished Becker paper and an unnamed NeurIPS submission

**Questions:**

If the probability of a counterfactual formula in CBNs is defined using causal models, could one not use Pearl's method to calculate the probability in the causal model without using (f)ccccs?

>p2 footnote: p(umide).
that should be p(u|e) ?

Does that footnote really describe what Pearl intended? Perhaps he meant that x,y, and e have to be disjoint variables when calculating P(Y_x = y | e)

What probability do you obtain in this example, if you calculate it according to your definition of counterfactual probability?

>4 159 for the exogenous variables

are those not the endogenous variables?

>4 161 , u)_Y ..

should that be , u_Y..) ?

(there are also parenthesis missing in p9 401 and elsewhere)

>5 199 In fact, in another NeurIPS submission (with a different set of authors)

are the authors also independent?

>5 213 the probability that the conditional events Y1 = 0 | X1 = 0 and Y1 = 1 | X = 1  hold simultaneously.

would that not be  [X1 ← 0](Y1 = 0) ∧ [X1 ← 1](Y1 = 1) ?

or is it the same? why would it be the same?

>7 316  all the cpts in M get the right probability in M

"in M" twice? Is on of them supposed to be "M'"?


>8 343 (at least, as long as all settings of the parents of a variable in the relevant entries of the cpt have positive probability).

and there are no unobserved variables?

>p11 section B

there is a lot of text here in the appendix which is hardly referenced by the main text.

are the results here less important?

>p13 557 in S. .. φ_S

what is S and φ_S? is that S_c? and then φ_c?

>p15 593 ψ_ic = Pr_M(/\..)

{1,..,s} that is {1,..,s_i} ?

ψ_ic is the formula /\...., not the Pr_M ?


>p15 595ff c_Pa(..)

c_Pa(Z) is the c_Z\in T_Z? That is only defined for Pa(Z) \in \cal Z, i.e. Pa(Z) != Z_{ij}.
Do you prove somewhere that there is no Z_{ij} with Pa(Z) = Z_{ij}?

>p16 617 "in S"

what is S?

**Limitations:**

yes

---

> ### Author Rebuttal · Authors · 2024-08-01
>
> We thank the reviewer for several insightful and detailed comments
> (even from the appendix!).
>
> p. 2: footnote: This should indeed be p(u \mid e). Thanks! According
> to our reading of Pearl, the procedure as we've described it is
> Pearl's.  He does not require disjointness.  Our method, even without
> independence assumptions, gives p(X=1|X=1) = 1 (as it should).
>
> p. 4, lines 159, 161, p. 9, 401: thanks for spotting these typos
>
> p. 5, l. 199: there is one common author in the papers
>
> p. 5, line 213: Technically,  Y1 = 0 | X1 = 0 and Y1 = 1 | X = 1 is an
> event in a CBN, while [X1 ← 0](Y1 = 0) ∧ [X1 ← 1](Y1 = 1) is an
> event in a functional causal model.  According to the mapping that we
> define, the former event holds in a CBN M iff the latter event holds
> in the causal model M' that it is M is mapped to (according to our
> mapping).  But, formally, they are different events.
>
> p. 7, line 316: indeed the second M should be M'
>
> p. 8, line 343: Technically, the statement here is true even if there
> are unobserved variables.  But if there are such variables, then it
> seems unreasonable to identify the probability of a formula \phi in a
> CBN M with the probability of \phi in a causal model i-compatible with M.
>
> p. 11: Section B consists of background material needed to prove the
> bound on the probability of necessity and sufficiency under our
> independence assumptions.  Since Pearl devotes a lot of discussion to
> these probabilities in his book, we thought that the comparison would
> be useful to the reader.  We will discuss this in more detail in the
> main text in the full paper.  It's not so much that these results are
> less important, but we were short of space, and we felt that they were
> quite as necessary for telling our story.
>
> p. 13, l. 557: Indeed, these subscripts were somewhat garbled. Thanks for
> catching this!
>
> p. 15, l. 593: Another good catch!
>
> p. 15, l. 595: Good catch, we will fix this.
>
> p. 16 line 617: Again, this should be S_c.

---

### Decision · Program_Chairs · 2024-09-25

**Decision:**

Accept (poster)

**Comment:**

The paper shows that interventional formulas of Causal Bayesian Netowrks are identifiable, assuming independence-of-mechanism (Pearl's concept of autonomy). The reviewers agree that the problem is important and quite interesting. Specifically, as one reviewer puts it, the paper builds off recent work showing that Pearl's proposed approach for calculating conditional probabilities involving interventions is incorrect. Using Pearl's concept of autonomy, the papers then build a series of formalisms to calculate arbitrary conditional probabilities in causal Bayesian networks and show how these formalisms relate CBNs and SCMs. This is important. As one reviewer points out, the independence-of-mechanism assumption is quite strong and should be even better justified. However, the authors addressed the issue well, admitting downsides but also providing interesting examples. Moreover, the link to Pearl's work is clear. So over all, I am with the authors and the other reviewers, and recommend to accept.